# DHEvo: Data-Algorithm Based Heuristic Evolution for Generalizable MILP Solving

## Abstract

Primal heuristics are crucial for accelerating the solving process of mixed integer programming (MILP) problems. While large language models (LLMs) have shown great promise in generating effective heuristics, existing methods often fail to generalize across instances within the same problem class, where we define a problem class as a set of MILP instances derived from the same mathematical model. This limitation arises because MILP instances within the same class can exhibit substantial structural and distributional heterogeneity. However, existing methods treat instances uniformly, averaging performance over limited samples and yielding heuristics that lack generalization. To address this, we propose DHEvo, a data-algorithm co-evolution framework that jointly evolves representative instances and tailored heuristics integrated into the open-source solver SCIP. DHEvo employs an LLM-based multi-agent system to generate and refine data-algorithm pairs iteratively, guided by fitness feedback. Experiments on diverse MILP benchmarks show that DHEvo significantly outperforms state-of-the-art hand-crafted, learning-based, and LLM-based methods in solution quality and generalization.

## 1 Introduction

Mixed-integer linear programming (MILP) is of central importance in combinatorial optimization, operations research, and computer science. It has been widely applied to a broad range of real-world problems, including supply chain optimization (Liu et al., 2008; Jeong et al., 2019; Jokinen et al., 2015), hardware design (Ma et al., 2019; Hafer, 1991), production scheduling (Chen, 2010; Caumond et al., 2009; Superchi et al., 2024), and energy management (Chang et al., 2004; Kassab et al., 2024; Zare et al., 2024). An MILP problem is often defined by numerous parameters, such as cost coefficients, constraints, and bounds. These can all be mathematically represented as:

$$z^\dagger := \min_{x \in P^\dagger} c^\top x, \quad P^\dagger = \left\{ x \in \mathbb{R}^n \mid Ax < b, \underline{\pi} \le x \le \overline{\pi}, x_j \in \mathbb{Z} \; \forall j \in \mathcal{I} \right\},$$

where $M^\dagger := (c, P^\dagger)$, $A \in \mathbb{R}^{m \times n}$, $b \in \mathbb{R}^m$, $c, x \in \mathbb{R}^n$, $\underline{\pi}, \overline{\pi} \in \mathbb{R}^n_\infty$, and $\mathcal{I} \subseteq \{1, \dots, n\}$ indexes the integer-constrained variables.

In practice, instances derived from the same application domain or model template can exhibit substantial variation in structure, constraint tightness, and feature distribution, leading to large intra-class diversity. Therefore, well-designed primal heuristics must not only contribute to accelerating the solving process but also generalize well across instances within the same problem class (Ong & Moore, 1984; Balas et al., 2004; Berthold, 2006; Wallace, 2010; Witzig & Gleixner, 2021).

Current advanced approaches to automated heuristic design leverage a combination of large language models (LLMs) and evolutionary computation (EC) to generate heuristic algorithms. This synergy (Liu et al., 2024b) has driven notable progress across domains including combinatorial optimization (Zhang et al., 2024c; Liu et al., 2024a), mathematical problem solving (Romera-Paredes et al., 2024; van Stein & Bäck, 2024), decision-making (Makatura et al., 2023; Wu et al., 2024), and MILP problems (Zhou et al., 2024; Ye et al., 2025; Li et al., 2024a).

Despite these advances, existing approaches exhibit a fundamental limitation: they typically apply the same treatment to all problem instances, thereby disregarding structural heterogeneity within a problem class. This oversimplified assumption hinders LLM-based evolutionary frameworks from

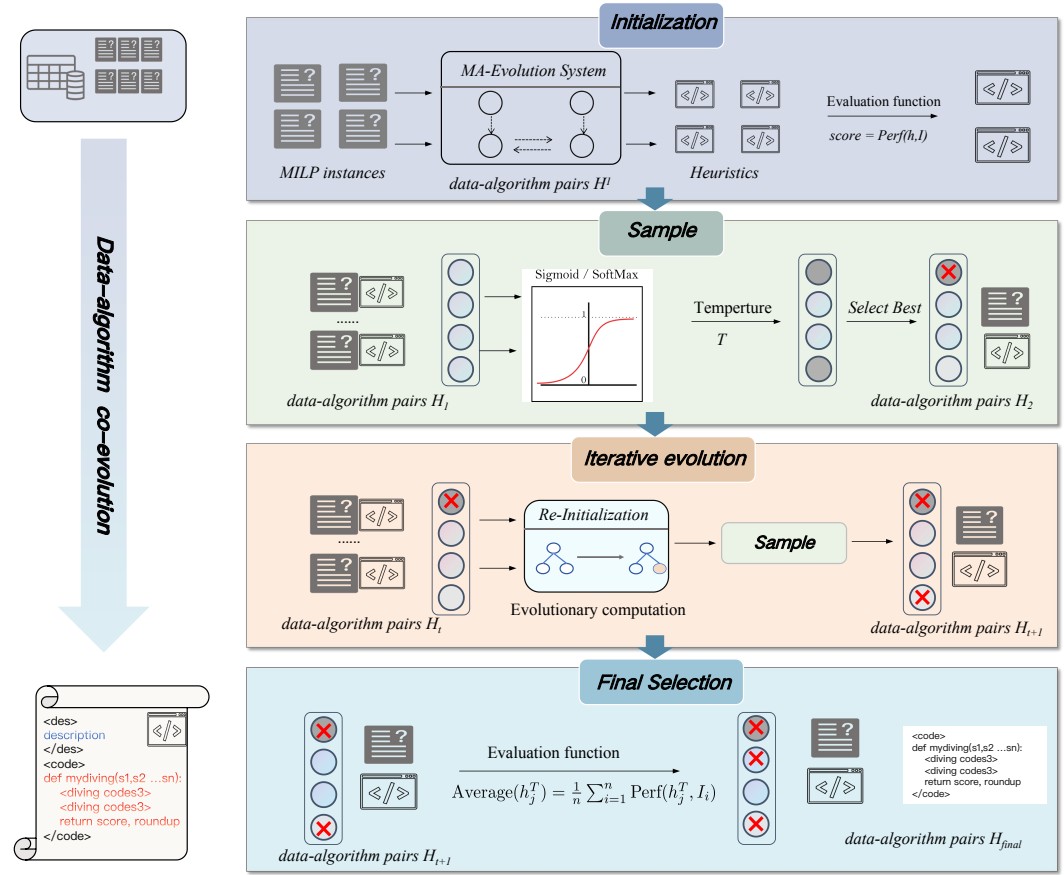

Figure 1: Illustration of data-algorithm co-evolution framework (DHEvo).

capturing representative structural patterns, resulting in heuristics that may demonstrate strong performance on specific training instances yet lack robustness and generalization across the broader instance distribution.

To address this issue, we propose a data-algorithm co-evolution framework (DHEvo) that generates generalizable algorithms by iteratively evolving both the MILP instances and the algorithms. We start by randomly sampling instances from a domain-specific dataset and developing an LLM-based multi-agent evolution system (MA-Evolution System) to create initial data-algorithm pairs. Inspired by insights from few-shot and curriculum learning (Ren et al., 2018; Sato et al., 2019; Bengio et al., 2009b), we select the pairs with the highest fitness (measured by relative primal gap) as the initial population for further evolution. Through our analysis of the instances (Section 3.2), we find that high-fitness pairs are more likely to encode transferable solving patterns, thereby enhancing generalization across instances within the same problem class. Then, the evolutionary process above iterates over generations, gradually refining the population toward the most representative data-algorithm pair. In summary, our contributions are as follows:

- We propose a unified data-algorithm co-evolution framework to evolve both instances and algorithms for the automatic design of heuristics. It enables better approximation of the instance distribution and increases the representational capacity of the learned heuristics, leading to improved generalization.

- We present a co-evolutionary solution for MILP tasks by instantiating the data-algorithm co-evolution paradigm through a multi-agent evolution system. Through continuous agent interaction and competition, the system fosters the emergence of diverse and adaptive heuristic strategies.

- Extensive experiments show that our method significantly improves the generalization of diving heuristics and delivers substantial performance gains across multiple MILP datasets.

## 2 BACKGROUND AND RELATED WORKS

### 2.1 BRANCH&BOUND AND DIVING HEURISTIC

A common method for solving MILP problems is Branch-and-Bound (B&B) (Land & Doig, 2009), which recursively builds a search tree by branching on fractional variables in the LP relaxation and pruning subproblems using objective bounds. Although B&B provides an exact framework, it remains computationally expensive for large-scale problems. To accelerate the search, solvers often incorporate primal heuristics such as diving, which conducts a depth-first search by iteratively rounding variables and re-solving LP relaxations until a feasible solution is found or infeasibility is detected. Existing diving heuristics, however, typically rely on manual design and expert tuning, limiting their adaptability. In contrast, our approach employs evolutionary computation to automatically generate problem-specific diving strategies, thereby enhancing flexibility and reducing dependence on expert knowledge. Empirical results show that this automated approach significantly improves primal gap progression across diverse benchmark datasets.

### 2.2 LLM FOR EVOLUTIONARY COMPUTATION

Evolutionary computation (Bäck et al., 1997) is a widely used method for solving optimization problems inspired by natural evolution. In recent years, the capabilities of large language models have advanced significantly (Naveed et al., 2023), and their integration with evolutionary computation has been explored for automated heuristic design (Liu et al., 2024b; Zhang et al., 2024c; Wu et al., 2024). For example, Funsearch (Romera-Paredes et al., 2024) combines LLMs with evolutionary frameworks to tackle mathematical problems, achieving superior results on the cap set and admissible set problems. EoH (Liu et al., 2024a) further integrates reasoning traces with executable code to generate more effective algorithms, achieving promising results on problems such as online bin packing. LLM4Solver (Zhou et al., 2024) integrates evolutionary search with LLMs to design heuristics for mixed-integer linear programming, improving solver efficiency across diverse datasets. Ye et al. (Ye et al., 2025) introduce a dual-layer self-evolving LLM agent for MILP, which automatically generates effective neighborhood selection strategies for large neighborhood search and generalizes from small-scale to large-scale instances.

However, current methods typically operate within a limited set of specific instances, limiting the ability of LLMs to capture the shared structural characteristics of the problem class. As a result, the generated algorithms perform well on similar instances but generalize poorly to broader problem variations. In contrast, our method iteratively selects representative instances during the evolutionary process, promoting the discovery of structural patterns that enhance generalization.

## 3 METHOD

### 3.1 PROBLEM FORMULATION

Instances within a single MILP problem class may exhibit substantial heterogeneity in distributions, constraints, and structural properties, while often retaining common characteristics such as constraint types, variable bounds, or recurring patterns in the objective function. Figure 2 presents a visualization of 17 representative features across four combinatorial optimization datasets. Therefore, systematically capturing such shared features is essential for the design of effective heuristics.

Conventional evolutionary methods typically evaluate heuristics by averaging performance over a small set of randomly sampled instances, implicitly assuming all instances are equally representative. In MILP, this assumption rarely holds due to high structural variability, resulting in a large performance variance over a broader instance set. To address this, our framework explicitly optimizes heuristics to achieve high expected performance while minimizing performance variance. Formally, the objective can be written as:

$$\min_{Q_p} R_{\mathcal{D}}(Q_p) = \mathbb{E}_{G \sim \mathcal{D}}[\ell(Q_p; G)], \tag{1}$$

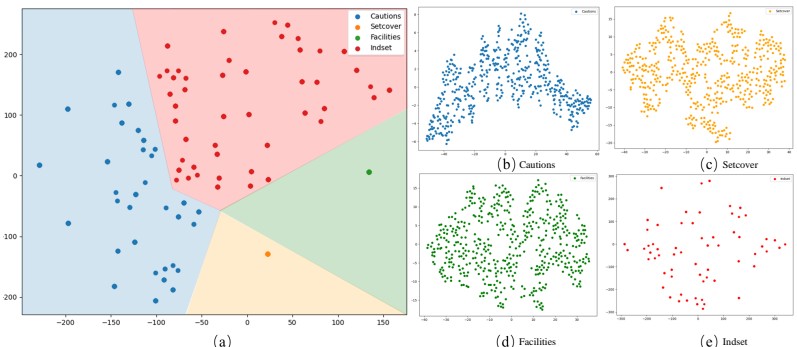

Figure 2: The visualization of instance features via t-SNE.

where $R_{\mathcal{D}}(Q_p)$ denotes the expected loss of heuristic $Q_p$ over the full MILP instance distribution $\mathcal{D}$, and $\ell(Q_p; G)$ measures the performance of $Q_p$ on a specific instance $G \in \mathcal{D}$.

### 3.2 THEORETICAL MOTIVATION: LEARNING FROM REPRESENTATIVE INSTANCES

**Insight** Motivated by few-shot learning (Jiang et al., 2015; Ren et al., 2018; Sato et al., 2019) and curriculum learning (Bengio et al., 2009a; Soviany et al., 2022; Portelas et al., 2020), extensive research (Wu et al., 2017; Akbari et al., 2021; Jiang et al., 2019) has shown that starting with "simple" or "representative" samples in complex datasets often enhances both learning efficiency and generalization. A similar phenomenon arises in the context of MILP optimization. Selecting structurally representative instances not only facilitates the discovery of effective heuristics but also improves their ability to generalize across the full problem class. Here, we provide a theoretical analysis that formalizes this intuition and motivates the design of our co-evolution framework. The detailed theoretical proofs are provided in Appendix 8.6.

**Definition 1** *MILP instance space and single diving operator loss.* Let $\mathcal{X}$ denote the space of MILP instances, and let $Q$ denote a single diving operator, which applies a rounding or branching decision to some subset of variables in an instance $G \in \mathcal{X}$. Given a finite training sample $S = \{G_1, \ldots, G_n\} \overset{iid}{\sim} \mathcal{D}$, the empirical risk is $\hat{R}_S(Q_p) := \frac{1}{n} \sum_{j=1}^{n} \ell(Q_p; G_j)$.

**Definition 2** *Complex diving heuristics as mixtures of atomic operators.* A diving operator is generally a decision rule over multiple variables. Let $\mathcal{H} = \{H_1, \ldots, H_k\}$ denote a finite set of *atomic* diving operators. Any complex diving heuristic $Q_p$ can be expressed as a convex combination of atomic operators: $Q_p := \sum_{i=1}^{k} p_i H_i$, where $p$ is a probability vector. The corresponding loss of $Q_p$ on an instance $G$ is $\ell(Q_p; G) := \sum_{i=1}^{k} p_i \ell(H_i; G)$, and the induced function class is

$$\mathcal{Q}_{\text{conv}} := \left\{ Q_p = \sum_{i=1}^{k} p_i H_i \ \Big| \ p \in \Delta^{k-1} \right\}, \quad \Delta^{k-1} := \left\{ p \in \mathbb{R}^k \ \Big| \ p_i \geq 0, \ \sum_{i=1}^{k} p_i = 1 \right\}.. \quad (2)$$

**Theorem 1** *Rademacher complexity of convex combinations.* Let $\sigma_j$ be independent Rademacher variables. The empirical Rademacher complexity of $\mathcal{Q}_{\text{conv}}$ on $S$ is

$$\hat{\mathfrak{R}}_S(\mathcal{Q}_{\text{conv}}) := \mathbb{E}_{\sigma} \left[ \sup_{Q_p \in \mathcal{Q}_{\text{conv}}} \frac{1}{n} \sum_{j=1}^{n} \sigma_j \ell(Q_p; G_j) \ \Big| \ S \right] = \hat{\mathfrak{R}}_S(\mathcal{H}). \quad (3)$$

**Remark 1** Theory 1 establishes that the Rademacher complexity of a function class formed by convex combinations of atomic diving operators is identical to that of the atomic operators themselves. Therefore, constructing complex heuristics from simple atomic operators preserves the original generalization capacity.

**Theorem 2** *Uniform generalization bound for mixtures of atomic operators.* Given a training sample $S$, the empirical risk is $\hat{R}_S(Q_p) = \frac{1}{n} \sum_{j=1}^{n} \ell(Q_p; G_j)$ and the expected risk is $R_{\mathcal{D}}(Q_p) = \mathbb{E}_{G \sim \mathcal{D}}[\ell(Q_p; G)]$. Then for any $\delta \in (0, 1)$, with probability at least $1 - \delta$ over $S$, simultaneously for

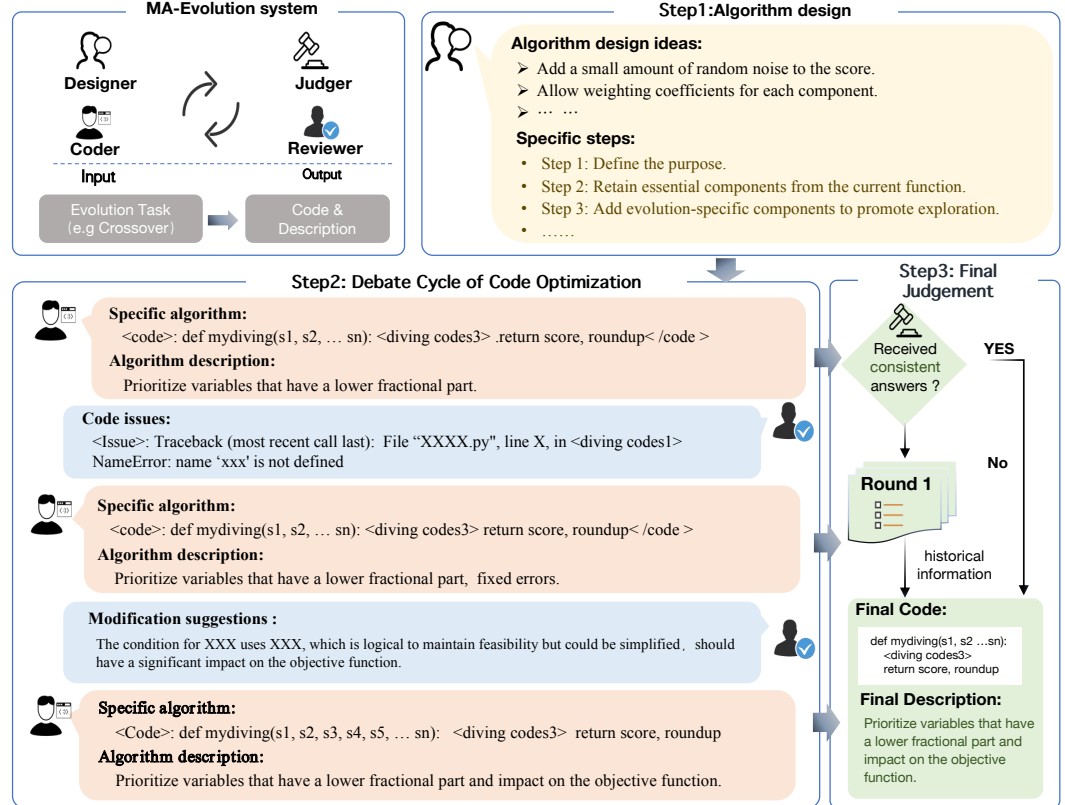

Figure 3: Illustration of the MA-Evolution System.

all $Q_p \in \mathcal{Q}_{\text{conv}}$:

$$R_{\mathcal{D}}(Q_p) \leq \hat{R}_S(Q_p) + 2B\sqrt{\frac{2\ln k}{n}} + B\sqrt{\frac{\ln(2/\delta)}{2n}}. \tag{4}$$

**Remark 2** Theorem 2 indicates that complex heuristics retain the same generalization bound as their atomic components. Hence, training on structurally representative, high-scoring instances is justified, as repeated optimization over such instances ensures guaranteed generalization performance.

### 3.3 DATA-ALGORITHM BASED HEURISTIC EVOLUTION FRAMEWORK

As illustrated in Figure 1, our framework adopts a structured evolutionary process that tightly couples instance selection with heuristic generation and optimization.

Initially, the MA-Evolution System generates a unique instance–heuristic pair for each sampled MILP instance, establishing an initial population of candidate algorithms tied to specific instances. Each generated heuristic is then evaluated on its corresponding instance, and a temperature-controlled selection strategy is applied to choose high-fitness instance–heuristic pairs. In general, heuristics with higher fitness scores correspond to instances with simpler structural characteristics. Subsequently, heuristics with low performance and their associated challenging instances are discarded. The remaining high fitness heuristics are then evolved further on the selected representative instances in the next generation. This process of generating, evaluating, selecting, and re-initializing instance–heuristic pairs is iterated over multiple generations. By repeatedly focusing on structurally representative and high-performing instances, the framework achieves co-evolution of instances and heuristics, ultimately producing algorithms with strong instance-level performance and reliable generalization across the broader problem class. Appendix 8.5 describes the detailed procedure of our method.

### 3.4 Framework implementation

**Evolution operation** Our evolutionary framework consists of four main operations: initialization, crossover, mutation, and parent selection. As shown in Figure 3, initialization, crossover, and mutation are implemented through sophisticated prompts to generate candidate individuals. We leverage the MA-Evolution System to perform both crossover and mutation, enabling more targeted and problem-aware generation of new individuals. Specific prompts are used only in the first generation to create the initial population, while subsequent generations reuse high-quality algorithms obtained from previous iterations. During crossover, parent heuristics are combined to form new candidate algorithms, and mutation introduces small variations to explore neighboring solutions. To balance exploration and exploitation in parent selection, we adopt fitness-proportional selection (Zhou et al., 2019), assigning selection probabilities to individuals based on their fitness scores.

**MA-Evolution System** To generate high-quality heuristics, we propose a multi-agent evolution system inspired by multi-agent systems (Liang et al., 2023; Chan et al., 2023; Zhang et al., 2024a; Li et al., 2024b). As shown in Figure 3, the process includes three stages. In the first stage, the *Designer* agent receives the MILP task context, existing code, and the specified evolutionary operation. It produces a high-level design plan and procedural outline for a new heuristic. In the second stage, the *Coder* agent implements the algorithm based on the Designer's plan. The *Reviewer* agent then checks the code by compiling it and performing logical analysis, providing feedback and suggestions. Then, the *Coder* and *Reviewer* iteratively improve the code through several rounds of interaction. In the final stage, if no consensus is reached, the *Judge* agent reviews the full interaction history and feedback, and makes a final decision on the output code and its description.

**Prompt engineering** Our prompt engineering is guided by three fundamental principles: (1) Evolution-Specific Alignment, where operations intrinsic to evolutionary computation are explicitly embedded; (2) Role-Based Specialization, which defines the designated function of the LLM within the MA-Evolution System; and (3) Problem Contextualization, incorporating specific contextual information about the MILP task. By strictly adhering to these core elements and avoiding unnecessary complexity, our framework maintains a lightweight design that facilitates seamless integration with other algorithm generation methods. The full prompt information is presented in the Appendix 8.7.

## 4 Experiments

### 4.1 Experimental settings

To demonstrate the superiority of our method in the diving task, we conduct two sets of experiments across six MILP datasets. (1) The first set of experiments is designed to study the diving performance of our method and compare it against existing diving heuristics. (2) To evaluate the efficiency improvement brought by our generated diving heuristics, we evaluate our method on combinatorial and large-scale real-world datasets. Experimental details are described in Appendix 8.4. (3) We extend our evaluation to classic combinatorial optimization problems to demonstrate the broad applicability of our framework Appendix 8.9. (4) Finally, we validate the generalization capability of our method across different problem classes Appendix 8.8.

### 4.2 Experiments on the Quality of Diving Heuristics

**Experimental setup** To evaluate the performance of the generated diving algorithms, we conduct two sets of experiments using the relative primal gap (Equation 7) as the primary metric, which measures the difference between the incumbent solution and the optimum. (1) We compare our method against a comprehensive set of existing diving heuristics, including human-designed, learning-based, and LLM-generated approaches. Specifically, we evaluate a total of 11 publicly available methods across four combinatorial optimization problems: cauctions, setcover, facilities, and indset. These baselines consist of six human-designed heuristics implemented in the open-source solver SCIP, the state-of-the-art learning-based GNN method L2DIVE (Paulus & Krause, 2023), and five LLM-generated heuristics: LLM4Solver (Zhou et al., 2024), FunSearch (Romera-Paredes et al., 2024), EoH (Liu et al., 2024a), ReEvo (Bergman et al., 2016) and HillClimb (Zhang et al., 2024b). (2) To further assess the superiority of our generated algorithms, we compare them against the mainstream

Table 1: The standard error and average relative primal gap (%) of different diving heuristics. The results compare our method with other LLM-based evolutionary approaches, as well as seven human-designed heuristics and the learning-based SOTA baseline.

| Category | Method | Cautions | Facilities | Setcover | Indset |
|---|---|---|---|---|---|
| LLM-based Evolution | **DHEvo(Ours)** | **1.92 (2.45)** | **0.70 (1.40)** | 9.74 (**7.35**) | 1.07 (1.20) |
| | LLM4Solver | 2.50 (3.50) | 0.85 (1.42) | 18.33 (19.26) | 1.13 (1.15) |
| | Funsearch | 3.04 (7.35) | 1.18 (3.06) | 77.99 (83.89) | 1.61 (3.75) |
| | HillClimb | 6.10 (60.30) | 0.75 (1.40) | 81.55 (343.17) | 1.61 (3.75) |
| | ReEvo | 6.11 (59.62) | 0.78 (1.21) | **7.82** (8.43) | 1.10 (**1.05**) |
| | EoH | 3.15 (3.15) | 0.80 (1.47) | 20.39 (19.70) | **0.92** (1.06) |
| Hand-crafted Heuristics | Coeficient | 23.67 (2.14) | 3.20 (3.76) | 68.58 (345.99) | 4.23 (14.42) |
| | Distributional | 47.80 (71.56) | 1.46 (2.12) | 75.79 (325.90) | 2.57 (10.59) |
| | Farkas | 23.32 (0.89) | 1.04 (1.64) | 8.13 (8.22) | - |
| | Pseudocost | 22.51 (2.30) | 1.06 (1.23) | 23.56 (30.31) | 3.31 (2.98) |
| | Linesearch | 22.95 (0.90) | 13.80 (10.94) | 68.59 (346.00) | 3.31 (3.10) |
| | Vectorlength | 42.93 (83.57) | 13.93 (10.61) | 68.59 (346.01) | 8.89 (7.61) |
| Learning-based | L2DIVE | 2.60 | 0.71 | 3.58 | 1.37 |

Table 2: The standard error and average relative primal gap (%) on four MILP datasets. Results are averaged over 100 new challenging instances per dataset, each on average over 4× harder than those in Table 1.

| Method | Cautions | Facilities | Setcover | Indset |
|---|---|---|---|---|
| Ours + SCIP | **1.22(2.66)** | **0.56(0.59)** | 3.79(2.80) | **0.51(0.63)** |
| Gurobi | 2.06(3.50) | 1.34(1.78) | **3.35(1.09)** | 1.93(3.41) |
| Tuned SCIP | 1.49(3.27) | 0.80(0.81) | 3.93(2.94) | 0.80(3.22) |

solvers Gurobi and SCIP. To ensure a fair comparison across heuristics and eliminate the potential bias introduced by the solver itself, we restrict the evaluation to the root node only.

**Experimental results** For the first set of experiments on diving heuristic performance, we compare our method against established diving heuristics. As shown in Table 1, our method consistently achieves strong results across all datasets. In particular, on the indset dataset, our approach improves over the best manually designed heuristic by 56.04%. Compared to other LLM-based algorithm design methods, our approach also achieves comparative performance. For example, on the setcover dataset, our method surpasses the best LLM-based baseline by 61.8%. More importantly, in terms of performance variance, our method achieves the lowest variance across all four datasets except for setcover. Notably, on the cautions and facilities datasets, our method achieved optimal results in both variance and mean.These results demonstrate the effectiveness and robustness of our approach in generating high-quality heuristics for diverse combinatorial optimization problems.

Secondly, we compare our generated heuristics with the primal heuristics embedded in state-of-the-art MILP solvers. As shown in Table 2, our method demonstrates highly competitive performance. In particular, the improvements over one of the leading solvers, Gurobi, range from approximately 24% on the cauctions dataset to more than 80% on the indset dataset. Unfortunately, we cannot embed our diving heuristics directly into commercial solvers like Gurobi to perform evolutionary optimization. On setcover datasets, our method still shows a performance gap relative to Gurobi.

## 4.3 EXPERIMENTS ON SOLVING EFFICIENCY IN BRANCH AND BOUND

**Experimental setup** To evaluate the practical effectiveness of the generated diving heuristics, we integrate them into SCIP and conduct experiments on both combinatorial optimization datasets and

Table 3: Performance comparison of our method, EoH, default SCIP, and tuned SCIP. Each cell reports the solving time and (primal-dual integral).

| Method | Cautions | Facilities | Setcover | Indset |
|---|---|---|---|---|
| Default SCIP | 4.08 (55.87) | 301.20 (506.71) | 2.43 (117.65) | 21.07 (230.33) |
| Tuned SCIP | 2.73 (24.21) | 201.64 (553.15) | 2.33 (77.02) | 22.71 (167.43) |
| EoH | 2.62 (37.12) | 197.35 (504.56) | 2.76 (96.75) | 20.32 (151.34) |
| **DHEvo(Ours)** | **2.28 (23.42)** | **181.27 (490.43)** | **2.27 (75.88)** | **18.54 (146.39)** |

Table 4: A comparison of solving time and primal-dual integral across different methods in large-scale real-world applications.

| | LoadBalance | | NNVerify | | MIPLIB | |
| --- | --- | --- | --- | --- | --- | --- |
| | Time | PDI | Time | PDI | Time | PDI |
| Ours + SCIP | 3600 | 346980.53 | 72.42 | 5413.32 | 263.48 | 12101.11 |
| Scip | 3600 | 347597.70 | 669.15 | 38455.17 | 469.22 | 18127.57 |
| **Ours + Tuned SCIP** | **1800** | **7305.2** | **35.67** | **2744.21** | **117.67** | **5599.62** |
| Tuned SCIP | 1800 | 9881.29 | 137.19 | 8210.46 | 184.3 | 6339.64 |

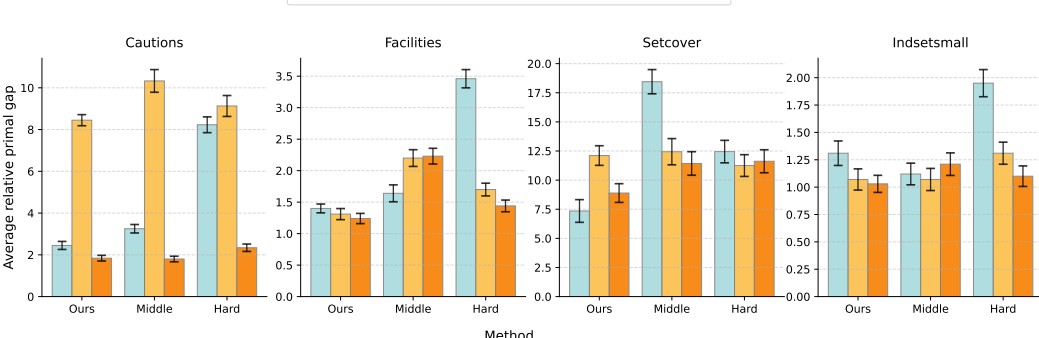

Figure 4: Ablation studies of different LLMs and data selection strategies on four problem classes.

large-scale real-world datasets, including LoadBalance (Gasse et al., 2022), MILPLIB (Gleixner et al., 2021), and NNVerify (Nair et al., 2020). Performance is assessed using solving time and the primal-dual integral, which together capture the solving efficiency.

**Experimental results** For the combinatorial optimization benchmarks, we compare our method to the default, tuned SCIP settings and EoH, as shown in Table 3. Results demonstrate that our approach not only improves solution quality but also leads to better solving efficiency. On the challenging facility dataset, our method outperforms the current state-of-the-art by 6.7% in solving time and 2.8% in primal-dual integral.

For the large-scale real-world datasets, we compare our method to the default and fine-tuned SCIP setting, as shown in Table 4. The experimental results demonstrate that our method achieves competitive improvements across all datasets. Under the fine-tuned setting, our method achieves a 26.1% improvement in the primal-dual integral on the LoadBalance dataset. On the NNVerify dataset, the fine-tuned approach more than doubles solving efficiency. For the MIPLIB dataset, our method improves solving efficiency by 36% and reduces the PDI by 12% compared to the default.

## 4.4 ABLATION STUDIES

To assess the contribution of each component in our framework, we conduct ablation studies on four combinatorial optimization datasets. Specifically, we evaluate: (i) whether high-fitness instances serve as structurally representative samples that guide the evolutionary process, (ii) the effectiveness of the data–algorithm co-evolution mechanism, (iii) the role of the multi-agent evolution system by comparing it with alternative evolutionary strategies, (iv) the robustness of our approach across different LLMs.

**Analysis on different data selection strategies** We conduct a detailed ablation study to investigate the correlation between simple instances and representative instances. Specifically, we evaluate three strategies for data selection: iteratively choosing simple instances, selecting instances of medium difficulty, and focusing on hard instances. As shown in the Figure 4, the results demonstrate that emphasizing simple instances yields more significant performance improvements compared to the other strategies. Notably, when using GPT-4o-mini, the evolutionary process guided by simple instances achieves a 24% higher improvement over the variant using medium difficulty instances and a 70% greater gain compared to the one focusing on hard instances. It indicates that simple data can serve as effective representatives for guiding heuristic evolution and enhancing generalization.

Table 5: Comparison of standard error and average relative primal gap on validation dataset, including DHEvo, its variant without co-evolution (DHEvo-OFF), the EoH baseline with co-evolution mechanism (EoH-DH), and the plain EoH framework.

| Method | Cautions | Facilities | Setcover | Indset |
|---|---|---|---|---|
| **DHEvo** | **2.15 (2.53)** | **0.83 (1.30)** | **9.74 (13.42)** | **1.01 (1.03)** |
| DHEvo-OFF | 2.33 (2.79) | 0.93 (1.45) | 10.8 (13.99) | 1.23 (1.11) |
| **EoH-DH** | **2.90 (5.60)** | **0.84 (1.47)** | **18.31 (17.48)** | **1.07 (1.14)** |
| EoH | 4.38 (6.15) | 1.96 (4.36) | 26.14 (28.89) | 1.36 (1.21) |

**Analysis on data-algorithm co-evolution** We evaluate the role of the co-evolution mechanism by removing it and using a uniform fitness evaluation over all training instances. Without this mechanism, performance variance increases and solution quality deteriorates across datasets. Specifically, when we remove the coevolution mechanism from DHEvo as shown in Table 5, the average relative primal gap increases by roughly 10% on each of the four combinatorial optimization datasets, demonstrating that uninformative or overly complex instances dominate the training process and harm generalization. When the EoH method is augmented with our co-evolution framework, it achieves significant improvements across all four datasets. This further demonstrates the effectiveness of our co-evolution mechanism in enhancing generalization and overall performance.

**Analysis on MA-Evolution System** To verify the effectiveness of the MA-Evolution System in generating higher-quality diversity generated individual algorithms, we conduct an ablation study by removing this system from our framework and comparing it with the original version in the setcover dataset. To evaluate the diversity of algorithms generated by the MA-Evolution System, inspired by diversity indicator metrics (Wineberg & Oppacher, 2003; Nikfarjam et al., 2021), we introduce a diversity index defined as $DI = H/\log_2 N$, where $H$ is the Shannon entropy of the score distribution over $N$ generated samples. A value closer to 1 indicates higher diversity among solutions.

As shown in the Table 6, the algorithms generated by the MA-Evolution System achieve significantly lower average primal gaps, improving by 12.4% compared to those without the MA-Evolution System. Additionally, they show a 15.8% improvement in the diversity index, demonstrating the superior diversity of the generated heuristics.

Table 6: Ablation study of the MA-Evolution system in terms of average primal gap (APG), diversity index (DI), and primal gap standard deviation (PGSD).

| Method | APG | DI | PGSD |
|---|---|---|---|
| MA-Evolution OFF | 9.14 | 0.76 | 8.75 |
| **MA-Evolution ON** | **8.00** | **0.88** | **4.78** |

**Analysis on different LLMs** We compare our method against several LLMs, including GPT-4o-mini, Qwen3-235B-A22B, and DeepSeek. All experiments are conducted under identical experimental settings to ensure a fair comparison. As shown in Figure 4, our approach consistently generates high-quality heuristics across all evaluated LLMs, demonstrating its robustness and generalizability irrespective of the underlying language model.

## 5 CONCLUSION

We present a novel data-algorithm co-evolution framework for solving MILP. By iteratively identifying the most representative instances and co-evolving heuristic algorithms based on them, our method significantly improves the generalization ability of the generated heuristics within the same problem class. Unlike traditional approaches that treat training data as static, our method selects representative instances during the evolutionary process, enabling the algorithm to generalize better across diverse problem distributions. We also introduce a multi-agent evolutionary system to improve generation quality and solution diversity. Experimental results show that our approach significantly outperforms existing human-designed, learning-based, and LLM-based baselines in both the primal gap and solving efficiency.

## 6 ETHICS STATEMENT

I have read the ICLR Code of Ethics and confirm that this work complies with all relevant ethical guidelines. I guarantee that the research was conducted responsibly, without harm to individuals or communities, and that all data usage adheres to applicable privacy and intellectual property standards.

## 7 ETHICS STATEMENT

We commit to full reproducibility of our results. All code, trained models, and datasets used in this work will be released under a permissive open-source license upon publication. Experimental details, hyperparameters, and evaluation protocols are provided in the appendix to ensure faithful replication.

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

# 8 APPENDIX

## 8.1 THE USE OF LARGE LANGUAGE MODELS

In our approach, the large language model (LLM) acts as an agent within the evolutionary computation framework. Rather than being used in isolation, the LLM generates candidate algorithms based on feedback from the evolutionary process. These algorithms are evaluated in real solver runs, and the performance results are fed back to guide the LLM in producing improved variants.

## 8.2 DIVING HEURISTICS

Diving heuristics are primal heuristics that iteratively fix variables based on LP relaxation solutions, simulating a depth-first search in the branch-and-bound tree. Given the LP relaxation of an MILP:

$$z_{LP}^\dagger := \min_{x \in P_{LP}^\dagger} c^\top x, \quad P_{LP}^\dagger = \left\{ x \in \mathbb{R}^n \mid Ax < b, \underline{\pi} \le x \le \overline{\pi} \right\},$$

the algorithm starts from an LP solution $\hat{x} \in P_{LP}^\dagger$ and incrementally fixes fractional variables $x_j \notin \mathbb{Z}$ to integer values. At each step, the feasible region is updated with new bound constraints, and the relaxed problem is re-solved. This process emulates a depth-first traversal of the search space, aiming to quickly construct a feasible integer solution. In general, a generic diving heuristic can be described by Algorithm 1. The only difference among various diving heuristics lies in the scoring function $s(\cdot)$, which determines the variable to round and the direction of rounding at each iteration.

Here are some diving heuristic algorithms included in SCIP.

**Coefficient.** This strategy selects a variable that has the smallest number of positive up-locks or down-locks. These locks represent how many constraints would prevent increasing or decreasing the variable, respectively. The variable is then fixed in the direction where fewer locks occur. If there is a tie between multiple variables, the method uses a secondary rule called fractional diving to break the tie.

---

**Algorithm 1** Generic Diving Heuristic

---

**Input:** MILP with relaxed feasible region $P^*$, LP solution $x^*$, maximum depth $d_{\max}$
**Output:** A set of feasible solutions $\mathcal{X}$ (if found)
**Require:** A scoring function $s$ for selecting branching variables and their rounding direction
1:  Initialize depth $d \leftarrow 1$, candidate set $\mathcal{C} \leftarrow \{j \in \mathcal{I} \mid x_j^* \notin \mathbb{Z}\}$
2:  **while** $d \leq d_{\max}$ **do**
3:      $j \leftarrow \arg\max_{i \in \mathcal{C}} s(x_i)$
4:      **if** round up **then**
5:          $l_j \leftarrow \lceil x_j^* \rceil$
6:      **else**
7:          $u_j \leftarrow \lfloor x_j^* \rfloor$
8:      **end if**
9:      $P^* \leftarrow P^* \cap \{l_j \leq x_j \leq u_j\}$
10:     **if** $P^*$ is infeasible **then**
11:         **break**
12:     **end if**
13:     $x^* \leftarrow \arg\min_{x \in P^*} c^\top x$
14:     **if** $x^*$ is roundable **then**
15:         $\mathcal{X} \leftarrow \mathcal{X} \cup \text{round}(x^*)$
16:     **end if**
17:     $d \leftarrow d + 1$
18:     Update candidate variable index set $\mathcal{C}$
19: **end while**

---

**Distribution.** This method is based on the empirical distribution of fractional values observed in historical solutions. It favors variables that are more frequently fractional in previous LP relaxations. The idea is that such variables are likely to remain fractional and therefore more useful for branching.

**Farkas.** This strategy tries to construct a Farkas proof to show the infeasibility of the current LP relaxation after branching. It selects the variable whose rounding, in the direction that improves the objective, is predicted to cause the largest gain. This prediction is based on LP dual information or inference from constraint violation. The method is designed to make branching decisions that quickly lead to pruning.

**Fractional.** This method selects the variable that is closest to an integer value, but still fractional. The measure used is $\left| x_j^* - \lfloor x_j^* + 0.5 \rfloor \right|$, which captures how far the variable's value is from the nearest integer. The selected variable is then rounded in the direction that brings it closest to an integer. This approach is simple and focuses on reducing the integrity gap.

**Linesearch.** This method traces a straight line (ray) from the root node LP solution to the current LP solution $x^*$. It identifies which integer hyperplane—either $x_j = \lfloor x_j^* \rfloor$ or $x_j = \lceil x_j^* \rceil$—is intersected first along this ray. The variable defining that hyperplane is selected for branching. This approach can be seen as a geometric way to decide which variable will influence the search path as soon as possible.

**Pseudocost.** This strategy uses historical data, called pseudocosts, to guide branching. For each variable, it records the average objective improvement caused by previous up- or down-branching decisions. The variable and branching direction with the highest expected improvement are selected. This method also considers the current fractionality of the variable to refine the choice. It is widely used due to its balance between accuracy and efficiency.

**Vectorlength.** This method is inspired by set-partitioning problems. It evaluates the trade-off between how much rounding a variable is expected to degrade the objective and how many constraints the variable appears in. The selected variable minimizes the ratio between the expected degradation and its constraint count. This helps prioritize variables that have a broad structural impact while limiting damage to the objective.

To guide our learned diving score function, we use variable-level features that are inspired by those employed in existing human-designed diving heuristics. These include 13 features in total, which are listed and described in Table 7.

Table 7: Description of the 13 input features used in the diving score function.

| Feature Name | Feature Description |
| --- | --- |
| *mayrounddown* | Boolean; indicates whether the variable can be rounded down while maintaining feasibility. |
| *mayroundup* | Boolean; indicates whether the variable can be rounded up while maintaining feasibility. |
| *candsfrac* | Float; fractional part of the variable's value in the LP relaxation, i.e., $|x_j^* - \lfloor x_j^* \rfloor|$. |
| *candsol* | Float; value of the variable in the current LP relaxation solution. |
| *nlocksdown* | Integer; number of down-locks, i.e., constraints that would be violated by decreasing the variable. |
| *nlocksup* | Integer; number of up-locks, i.e., constraints that would be violated by increasing the variable. |
| *obj* | Float; coefficient of the variable in the objective function. |
| *objnorm* | Float; Euclidean norm of the objective function coefficient vector. |
| *pscostdown* | Float; pseudocost for decreasing the variable's value. |
| *pscostup* | Float; pseudocost for increasing the variable's value. |
| *rootsolval* | Float; value of the variable in the LP relaxation at the root node. |
| *nNonz* | Integer; number of nonzero entries in the variable's column in the constraint matrix. |
| *isBinary* | Boolean; TRUE if the variable is binary, i.e., has domain $\{0, 1\}$. |

## 8.3 PERFORMANCE MEASUREMENT

To evaluate the performance of MILP solvers, we use several key performance metrics: Primal-Dual Gap, Primal-Dual Integral, and Primal Gap.

**Primal-Dual Gap** It is a widely used measure that quantifies the difference between the primal objective value and the dual objective value at any given time during the optimization process. It gives an indication of how close the current solution $\tilde{z}$ is to an optimal solution $\tilde{z}^*$. Mathematically, the Primal-Dual Gap is defined as:

$$\gamma_{pd}(\tilde{z}, \tilde{z}^*) = \begin{cases} \frac{|\tilde{z} - \tilde{z}^*|}{\max(|\tilde{z}|, |\tilde{z}^*|)} & \text{if } 0 < \tilde{z}, \tilde{z}^* < \infty, \\ 1 & \text{otherwise.} \end{cases} \tag{5}$$

**Primal-Dual Integral** While the primal-dual gap captures a snapshot at a particular time, the primal-dual integral evaluates the solver's progress over the entire solving process by aggregating the primal-dual gap over time. It is given by:

$$\gamma_{pdi}(t) = \int_0^t \gamma_{pd}(\tilde{z}(\tau), \tilde{z}^*(\tau)) \, d\tau, \tag{6}$$

where $\gamma_{pd}(\tilde{z}(\tau), \tilde{z}^*(\tau))$ represents the Primal-Dual Gap at time $\tau$.

**Primal Gap** It is used to evaluate the effectiveness of diving heuristics, which primarily aim to improve the primal performance by guiding the search toward better feasible solutions. The relative primal gap is defined as the absolute difference between the current objective value $\tilde{z}$ and the optimal solution $z^\dagger$, normalized by the objective value of the optimal solution. The formula for the primal gap is given by:

$$\gamma_p(\tilde{z}) = \frac{|\tilde{z} - z^\dagger|}{|z^\dagger|}, \tag{7}$$

where $z^\dagger$ is the objective value of the optimal solution obtained after presolving. In the case where $|z^\dagger| = 0$, we use the following modified primal gap:

$$\gamma_p'(\tilde{z}) = |\tilde{z} - z^\dagger|. \tag{8}$$

## 8.4 EXPERIMENTAL DETAILS

In all the experiments, we evaluate the performance of agents driven by GPT-4o mini across various tasks. We run all the experiments with three random seeds on Intel(R) Xeon(R) CPU E5-2667 v4 @ 3.20GHz and NVIDIA A100.

Table 8: Table 10: Used MIPLIB instance names

| | | | | |
|---|---|---|---|---|
| air05 | beasleyC3 | binkar10_1 | cod105 | dano3_3 |
| eil33-2 | hypothyroid-k1 | istanbul-no-cutoff | markshare_4_0 | mas76 |
| mc11 | mik-250-20-75-4 | n5-3 | neos-860300 | neos-957323 |
| neos-1445765 | nw04 | piperout-27 | pk1 | seymour1 |

*Note:* Since the code for L2DIVE is currently not open-source and specific hyperparameters are unavailable, we officially report the performance of L2DIVE based on its ratio to the best human-designed heuristic as presented in the original article. **SCIP settings** To construct our Tuned baseline, we incorporated domain knowledge and performed a randomized search over key diving-related parameters in SCIP 7.0.2. The primary parameters that govern the invocation of individual diving heuristics are *freq* and *freqofs*. These parameters determine when and how frequently a given diving heuristic is triggered during the branch-and-bound process. By adjusting their values, we can generate diverse solver behaviors that vary the timing and intensity of heuristic application. For each diving heuristic, we independently sampled its configuration by setting *freq* to one of four values with equal probability: $-1$ (disabled), $\lfloor 0.5 \times freq_{\text{default}} \rfloor$ (increased frequency), $freq_{\text{default}}$ (default frequency), or $\lfloor 2 \times freq_{\text{default}} \rfloor$ (reduced frequency). In parallel, we randomly set *freqofs* to either zero or its default value, also with equal probability. This approach allows us to sample a wide range of heuristic schedules while maintaining compatibility with established SCIP parameter semantics.

We evaluate our method on seven benchmark datasets, including four synthetic combinatorial optimization problems and three real-world MILP tasks. The datasets are widely used in prior work and include:

- *Setcover*: A classical combinatorial problem where the objective is to select a minimum number of subsets such that their union covers all elements. Instances are represented as binary matrices with rows corresponding to elements and columns to subsets. Easy instances have 500 rows and 1000 columns, while hard instances increase the size to 2000 rows and 1000 columns.

- *Cauctions*: A combinatorial auction problem where bidders submit bids on bundles of items, aiming to maximize total revenue without violating item availability constraints. Easy instances contain 100 items and 500 bids, while hard instances include 300 items and 1500 bids.

- *Facilities*: A capacitated facility location problem involving the selection of facility sites and the assignment of customers to minimize facility opening and service costs. Easy instances consist of 100 facilities and 100 customers, whereas hard instances have 100 facilities and 400 customers.

- *Indset*: The maximum independent set problem, which seeks the largest possible set of mutually non-adjacent vertices within a graph. Easy instances feature 500 nodes with an affinity of 4, and hard instances have 1500 nodes with the same affinity.

- *LoadBalance*: A server load balancing problem arising in distributed systems, modeled as an MILP.

- *NNVerify*: A verification problem for neural networks, where constraints encode input-output relationships that must be satisfied.

- *MIPLIB*: It contains a diverse collection of real-world and academic instances spanning various domains such as scheduling, network design, logistics, and combinatorial optimization. We selected 20 instances for experimental comparison.

The first experimental group is conducted on the four synthetic datasets, focusing on diving performance. The second group uses the three real-world datasets and synthetic datasets to demonstrate the effectiveness of our method in the practical solving process.

**Experiments on the Quality of Diving Heuristics** In this first set of experiments, we evaluate the quality of the learned heuristic algorithms in isolation by applying the diving heuristic only at the root node of each instance. All other solver components—such as branching rules, cutting planes, and primal heuristics—are disabled to ensure a controlled comparison. For fitness evaluation during

Table 9: Instance generation algorithms and detailed hyperparameters.

| Benchmark | Algorithm | Hyperparameters |
|---|---|---|
| Setcover | Balas & Ho (2009) | Easy: 500 rows, 1000 columns
Hard: 2000 rows, 1000 columns |
| Cauctions | Leyton-Brown et al. (2000) | Easy: 100 items for 500 bids
Hard: 500 items for 1500 bids |
| Facilities | Cornuéjols et al. (1991) | Easy: 100 facilities, 100 customers
Hard: 100 facilities, 400 customers |
| Indset | Bergman et al. (2016) | Easy: 500 nodes with affinity 4
Hard: 1000 nodes with affinity 4 |

evolution, we generate 50 training instances each for the setcover, cauctions, and indset datasets, and 25 for facilities. The evolved diving heuristics are then tested on 100 unseen instances per dataset. To ensure fairness, all LLM-based evolutionary methods are trained on the same dataset and use identical API interfaces. Furthermore, their prompts are carefully aligned with ours in terms of task context, including MILP-specific background and diving-related objectives, enabling a direct and equitable comparison.

In this second set of experiments, we integrate the evolved diving heuristic into SCIP and compare its performance against the default versions of SCIP and Gurobi on the same set of challenging instances. This comparison evaluates the practical benefit of incorporating our learned heuristic into a state-of-the-art solver. Compared to the initial benchmark, we increase the problem size and constraint density according to the parameter settings detailed in Table 9. Specifically, each instance has approximately 1000 variables and 2000 constraints for setcover, 1500 variables and 580 constraints for cauctions, 40100 variables and 40200 constraints for facility, and about 1000 variables and 4000 constraints for indset.

**Experiments on solving efficiency in branch and bound** On the combinatorial optimization datasets, we evaluate the solving efficiency of our method by comparing it against three baselines: the default SCIP solver, a tuned version of SCIP (with adjusted `freq` and `freqofs` parameters), and EoH. Experiments are conducted on the same four combinatorial optimization benchmark datasets. For each dataset, we randomly generate 1000 instances and select the 100 most challenging ones for evaluation. A time limit of $T_{\text{limit}} = 900$ seconds is imposed per instance, and performance is measured using the primal-dual integral.

To assess the performance of the proposed heuristic framework in realistic scenarios, we conduct experiments on three representative datasets: LoadBalance, MILPLIB, and NNVerify, which cover a broad range of MILP problem structures. Across all datasets, we adopt two standard performance metrics: the primal-dual integral, which captures convergence behavior and solution quality over time, and the solving time (T), which measures how quickly a feasible or optimal solution is found. For LoadBalance, we use 100 instances for validation and another 100 for testing, with $T_{\text{limit}} = 3600$ seconds as the standard setting and $T_{\text{limit}} = 1800$ seconds for additional robustness evaluation under tighter budgets. For MILPLIB, we select 20 relatively simple benchmark instances as a test set to evaluate generalization performance on classical MILP formulations; the instance names are listed in Table 8. For NNVerify, we evaluate on 100 testing instances derived from neural network verification problems, using a time limit of $T_{\text{limit}} = 900$ seconds and considering only instances successfully solved within the limit. To isolate the contribution of the learned diving heuristics, we perform all experiments under both `cut-selection` enabled and disabled configurations. In all settings, the heuristics are integrated into SCIP, and the best-performing variant is selected on the validation set based on either PDI or solving time before being applied to the testing set, mirroring realistic deployment scenarios.

## 8.5 IMPLEMENTION DETAILS

We first extend the SCIP solver by implementing a C-Python interface within its source code, enabling seamless communication between the solver and our learning framework. After recompiling SCIP with this extension, we integrate the learned diving heuristic—implemented as a Python

callable function into the solving process. At each node of the B&B tree, the heuristic receives a 13-dimensional feature vector describing the current variable and solution state. It then computes a score and a preferred rounding direction for each candidate variable. The variable with the highest score is selected for diving, and branching proceeds accordingly.

To evaluate the quality of each generated heuristic, we set a limit of one branch and bound node during SCIP's search. Given a problem instance $I_0$ with known optimal objective value $z^\dagger$, we execute the solver from the root node. When the generated diving heuristic is first invoked, we record the objective value $\tilde{z}$ of the best feasible solution found so far. The fitness score is then computed as the relative gap between $\tilde{z}$ and $z^\dagger$, defined as:

$$\text{Perf}(.) = \frac{|\tilde{z} - z^\dagger|}{|z^\dagger + \epsilon|}, \tag{9}$$

where $\epsilon$ is a small constant (e.g., $10^{-8}$) to prevent division by zero. A smaller gap indicates better early search performance, and thus higher fitness, guiding the evolutionary process toward heuristics that quickly identify high-quality feasible solutions.

Following the evaluation, we rank all generated heuristics based on their fitness scores across the corresponding instances. For each instance $I_0$, the heuristic that achieves the smallest $\text{Perf}(h, I)$ gap is considered superior. We then select the top 20% of algorithm-instance pairs with the best performance for the next evolutionary stage.

To maintain diversity and avoid premature convergence, we further apply a temperature-based sampling strategy. Instead of deterministic selection, we compute a probability distribution over the candidate pairs using a softmax function parameterized by a temperature $T$. This allows controlled stochasticity in selection, balancing exploitation of high-performing heuristics and exploration of potentially promising ones. The sampling probability for the $i$-th pair is defined as:

$$Sample(.) = \frac{\exp(-\text{Perf}(I_i)/T)}{\sum_j \exp(-\text{Perf}(I_j)/T)}, \tag{10}$$

where the negative sign ensures that lower performance gaps (better results) lead to higher probabilities, and the temperature $T > 0$ controls the sharpness of the distribution. Lower $T$ values favor exploitation, while higher values promote uniform exploration.

## 8.6 THEORETICAL ANALYSIS

**Theorem 1** *Rademacher complexity of convex combinations.*

*Let $\mathcal{Q}_{\text{conv}} = \{Q_p : p \in \Delta^{k-1}\}$ be the class of all convex mixtures of the atomic set $\mathcal{H}$. Then*

$$\hat{\mathfrak{R}}_S(\mathcal{Q}_{\text{conv}}) = \hat{\mathfrak{R}}_S(\mathcal{H}).$$

*In particular, forming convex combinations of the atomic operators does not increase the empirical Rademacher complexity.*

**Proof.** Compute the inner supremum in the definition of $\hat{\mathfrak{R}}_S(\mathcal{Q}_{\text{conv}})$:

$$\sup_{Q_p \in \mathcal{Q}_{\text{conv}}} \frac{1}{n} \sum_{j=1}^n \sigma_j \ell(Q_p; G_j) = \sup_{p \in \Delta^{k-1}} \frac{1}{n} \sum_{j=1}^n \sigma_j \sum_{i=1}^k p_i \ell(H_i; G_j).$$

Interchange summations (finite sums commute) and factor out the dependence on $p$:

$$= \sup_{p \in \Delta^{k-1}} \sum_{i=1}^k p_i \left( \frac{1}{n} \sum_{j=1}^n \sigma_j \ell(H_i; G_j) \right).$$

For fixed real numbers $a_i := \frac{1}{n} \sum_{j=1}^n \sigma_j \ell(H_i; G_j)$, the quantity $\sup_{p \in \Delta^{k-1}} \sum_i p_i a_i$ is the maximum of a linear functional over the simplex $\Delta^{k-1}$. A linear functional over a simplex achieves its maximum at an extreme point, i.e., at some standard basis vector. Hence

$$\sup_{p \in \Delta^{k-1}} \sum_{i=1}^k p_i a_i = \max_{1 \le i \le k} a_i = \sup_{H \in \mathcal{H}} \frac{1}{n} \sum_{j=1}^n \sigma_j \ell(H; G_j).$$

---

**Algorithm 2** DHEvo framework

---

**Require:** Problem distribution $\mathcal{D}$, population size $m$, number of instances $n$, top-$k$, total iterations $T$
**Ensure:** Final heuristic algorithm population $\mathcal{H}^{\text{final}}$

1: **Initialization:** Sample initial instance set $\mathcal{I}_0 \in \{I_1, \ldots, I_n\} \sim \mathcal{D}$
2: Generate initial algorithm population $\mathcal{H}_0 = \{h_1^{(0)}, \ldots, h_m^{(0)}\}$ via MA-Evolution System (MA-E)
3: $\mathcal{P}^0 = \{\}$
4: **for** each $h_j^{(0)} \in \mathcal{H}_0$ **do**
5:     Evaluate each algorithm: $f_j^0 = \text{Perf}(h_j^0, I_j)$
6:     $\mathcal{P}^0 \leftarrow \{f_j^0, I_j, h_j^0\}$
7: **end for**
8: **for** each $I_i \in \mathcal{I}_0$ **do**
9:     Generate $\mathcal{H}_1 = \{\text{MA-E}(I_j, h_j^0)\}_1^m$
10:     Evaluate each algorithm to obtain $f_j^1$ for each $h_j^1 \in \mathcal{H}_1$
11:     Update $\mathcal{P}^1 \leftarrow \{f_j^1, I_j, h_j^1\}$
12: **end for**
13: Let $\mathcal{P}^{t+1} \leftarrow$ the top-$k$% pairs from $\{(f_j^t, I_j, h_j^t)\}_{j=1}^{|\mathcal{P}^t|}$ ranked by $f_j^t$
14: **for** iteration $t = 2$ to $T$ **do**
15:     **Re-Initialization:**
16:     **for** each $(I_j^t, h_j^t) \in \mathcal{P}^t$ **do**
17:         Generate new candidates via prompt: $\mathcal{H}_t = \text{MA-E}(I_j^t, \text{Prompt}(h_j^{t-1}))$
18:         Evaluate each $h_j^t \in \mathcal{H}_t$
19:         Update $\mathcal{P}^{t+1} \leftarrow \text{top-}k\% \left(\mathcal{P}^t; f_j^t = \text{Perf}(h_j^t, I_j)\right)$
20:     **end for**
21: **end for**
22: **Final Selection:**
23: Compute $\text{Average}(h_j^T) = \frac{1}{n} \sum_{i=1}^n \text{Perf}(h_j^T, I_i)$
24: $\mathcal{H}_{\text{final}} = \underset{j}{\text{argmin}} \, \text{Average}(h_j^T)$
25: **return** $\mathcal{H}_{\text{final}}$

---

Taking expectation over the Rademacher variables $\sigma$ (and conditioning on $S$) preserves the equality, therefore

$$\hat{\mathfrak{R}}_S(\mathcal{Q}_{\text{conv}}) = \mathbb{E}_\sigma \left[ \sup_{H \in \mathcal{H}} \frac{1}{n} \sum_{j=1}^n \sigma_j \ell(H; G_j) \right] = \hat{\mathfrak{R}}_S(\mathcal{H}).$$

**Theorem 2** *Uniform generalization bound for mixtures. Assume $|\mathcal{H}| = k$ and $\ell(\cdot; \cdot) \in [0, B]$. For any $\delta \in (0, 1)$, with probability at least $1 - \delta$ over the draw of $S \sim \mathcal{D}^n$, the following holds simultaneously for all $Q_p \in \mathcal{Q}_{\text{conv}}$:*

$$R_\mathcal{D}(Q_p) \leq \hat{R}_S(Q_p) + 2B\sqrt{\frac{2\ln k}{n}} + B\sqrt{\frac{\ln(2/\delta)}{2n}}. \tag{11}$$

**Proof.** The proof proceeds in two steps: (i) bound the empirical Rademacher complexity of the finite atomic class $\mathcal{H}$ using Massart's lemma; (ii) apply a standard Rademacher-based uniform generalization inequality.

**Step (i) — Massart's lemma.** For a finite class $\mathcal{H}$ of cardinality $k$ whose function values lie in an interval of length at most $B$ (here $[0, B]$), Massart's lemma yields

$$\hat{\mathfrak{R}}_S(\mathcal{H}) \leq B\sqrt{\frac{2\ln k}{n}}.$$

Combining this with Theorem 1 gives the same bound for the convex hull:

$$\hat{\mathfrak{R}}_S(\mathcal{Q}_{\text{conv}}) = \hat{\mathfrak{R}}_S(\mathcal{H}) \leq B\sqrt{\frac{2\ln k}{n}}.$$

**Step (ii) — Rademacher generalization bound.** A standard Rademacher-based uniform deviation bound (for bounded functions) states that with probability at least $1 - \delta$ (over the draw of $S$), every

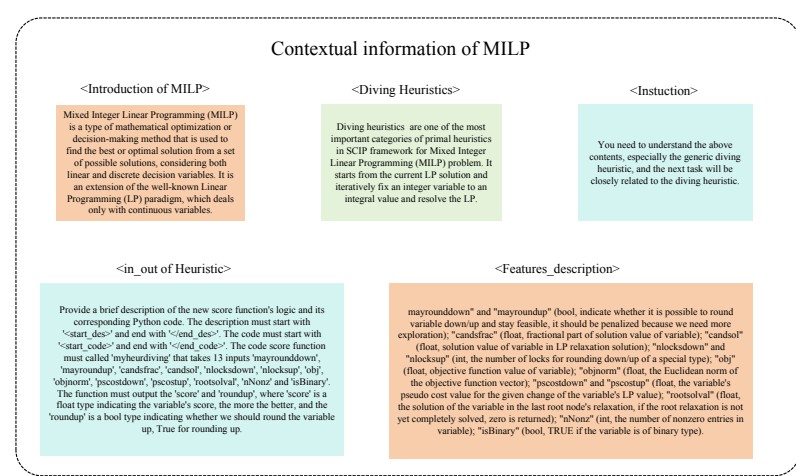

Figure 5: The prompts of contextual information of MILP.

function $f$ in a class $\mathcal{F}$ satisfies:

$$R(f) \leq \hat{R}_S(f) + 2\hat{\mathfrak{R}}_S(\mathcal{F}) + B\sqrt{\frac{\ln(2/\delta)}{2n}}.$$

Apply this inequality with $\mathcal{F} = \mathcal{Q}_{\text{conv}}$ and combine with the Massart bound above:

$$R_{\mathcal{D}}(Q_p) \leq \hat{R}_S(Q_p) + 2B\sqrt{\frac{2\ln k}{n}} + B\sqrt{\frac{\ln(2/\delta)}{2n}},$$

### 8.7 PROMPTS

**Prompts design** Our prompt design adopts a structured and modular format to effectively guide LLMs in performing evolutionary search within the multi-agent evolutionary framework. Each prompt is composed of three essential components, designed to provide the LLM with both domain-specific grounding and a clear operational goal.

As shown in Figure 5, background prompts contain *Introduction of MILP, Diving Heuristics, Instruction , in_out of Heuristic, Features description*. Together, they provide enough background knowledge of diving heuristics for the downstream tasks. Prompts in MA-Evolution System are modular and follow a structured template to ensure consistency across generations, as shown in Figure 6. At the core of each prompt are three elements: (1) the functional role of the agent, which defines the nature of the task (*e.g.*, proposing a new heuristic or reviewing existing code); (2) a formal or semi-formal description of the MILP problem to ground the response in the relevant optimization context; and (3) a specification of the evolutionary operation that informs the agent's goal in the current generation cycle.

Our evolution operation's prompt includes three main types: initialization, mutation, and crossover. Each type corresponds to a distinct stage in the evolutionary search process and is designed to guide LLMs in generating or improving heuristics for MILP diving.

*Initialization* The LLM is instructed to create a new scoring function from scratch. The function should assign a score and a rounding direction to each fractional variable, based only on the LP relaxation and objective function. This stage initializes the population with diverse and problem-aware heuristics.

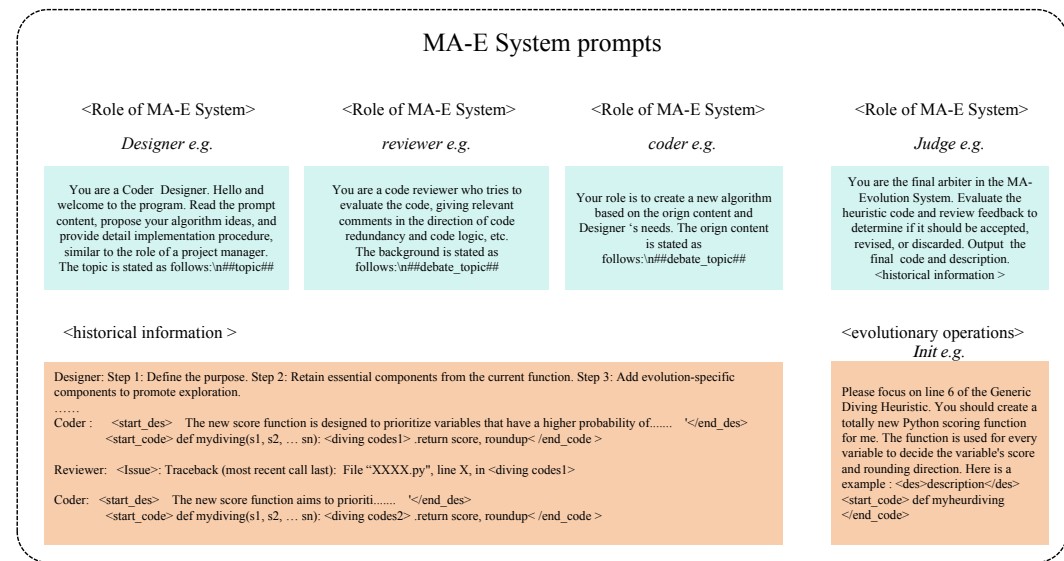

Figure 6: The prompts in the MA-Evolution System.

*Example prompt:* Please create a new Python scoring function for a Generic Diving Heuristic. The function should assign a score and rounding direction to each fractional variable, using only information from the LP relaxation and the objective function.

*Mutation* The LLM receives an existing scoring function and modifies it slightly. The modification should be meaningful and aimed at improving performance or exploring nearby variants in the heuristic space. This enables local search around known good solutions.

*Example prompt:* Please make a small but meaningful change that may improve performance or explore alternative behavior. Ensure the result is syntactically correct and remains within the MILP context.
Original function: [insert code]

*Crossover* The LLM combines two existing scoring functions into a new one. It should preserve useful components from both parents while ensuring the resulting function is coherent and consistent. This enables global search by recombining successful patterns.

*Example prompt:* You are creating a new heuristic by combining two existing ones. Please synthesize a scoring function that inherits effective components from both parents while maintaining logical consistency.
Heuristic A: [insert code]
Heuristic B: [insert code]

**Example** The following is an example of our method applied within DHEvo:

*Designer* You are a Coder Designer. Hello and welcome to the program. Read the prompt content, propose your algorithm ideas, and provide detailed implementation procedure, similar to the role of a project manager. The topic is stated as follows:Diving heuristics are one of the most important categories of primal heuristics in SCIP framework for Mixed Integer Linear Programming (MILP) problem. It starts from the current LP solution and iteratively fix an integer variable to an integral value and resolve the LP. You should create a totally new Python scoring function for me (different from the heuristics in the literature) to choose the fractional variable and corresponding rounding direction using the information of the LP relaxation and objective function. The function is used for every variable to decide the variable's score and rounding direction. Specifically, you have these features to use in the score function: "mayrounddown" and "mayroundup" (bool, indicate whether it is possible to round variable down/up and stay feasible, it should be penalized because we need more exploration); "candsfrac" (float, fractional part of solution value of variable); "candsol" (float, solu-

tion value of variable in LP relaxation solution); "nlocksdown" and "nlocksup" (int, the number of locks for rounding down/up of a special type); "obj" (float, objective function value of variable); "objnorm" (float, the Euclidean norm of the objective function vector); "pscostdown" and "pscostup" (float, the variable's pseudo cost value for the given change of the variable's LP value); "rootsolval" (float, the solution of the variable in the last root node's relaxation, if the root relaxation is not yet completely solved, zero is returned); "nNonz" (int, the number of nonzero entries in variable); "isBinary" (bool, TRUE if the variable is of binary type). Provide a brief description of the new score function's logic and its corresponding Python code. The description must start with 'start_des' and end with '/end_des'. The code must start with 'start_code' and end with '/end_code'. The code score function must call 'myheurdiving' that takes 13 inputs 'mayrounddown', 'mayroundup', 'candsfrac', 'candsol', 'nlocksdown', 'nlocksup', 'obj', 'objnorm', 'pscostdown', 'pscostup', 'rootsolval', 'nNonz', and 'isBinary'. The function must output the 'score' and 'roundup', where 'score' is a float type indicating the variable's score, the more the better, and the 'roundup' is a bool type indicating whether we should round the variable up, True for rounding up. Be creative and do not give additional explanations.

*Coder* Your role is to create a new algorithm based on the original content and the Designer's needs. The original content is stated as follows: Diving heuristics are one of the most important categories of primal heuristics in the SCIP framework for Mixed Integer Linear Programming (MILP) problems. It starts from the current LP solution and iteratively fixes an integer variable to an integral value and resolves the LP. You should create a new Python scoring function for me (different from the heuristics in the literature) to choose the fractional variable and corresponding rounding direction using the information of the LP relaxation and objective function. The function is used for every variable to decide the variable's score and rounding direction. Specifically, you have these features to use in the score function: "mayrounddown" and "mayroundup" (bool, indicate whether it is possible to round variable down/up and stay feasible, it should be penalized because we need more exploration); "candsfrac" (float, fractional part of solution value of variable); "candsol" (float, solution value of variable in LP relaxation solution); "nlocksdown" and "nlocksup" (int, the number of locks for rounding down/up of a special type); "obj" (float, objective function value of variable); "objnorm" (float, the Euclidean norm of the objective function vector); "pscostdown" and "pscostup" (float, the variable's pseudo cost value for the given change of the variable's LP value); "rootsolval" (float, the solution of the variable in the last root node's relaxation, if the root relaxation is not yet completely solved, zero is returned); "nNonz" (int, the number of nonzero entries in variable); "isBinary" (bool, TRUE if the variable is of binary type). Provide a brief description of the new score function's logic and its corresponding Python code. The description must start with 'start_des' and end with '/end_des'. The code must start with 'start_code' and end with '/end_code'. The code score function must call 'myheurdiving' that takes 13 inputs: 'mayrounddown', 'mayroundup', 'candsfrac', 'candsol', 'nlocksdown', 'nlocksup', 'obj', 'objnorm', 'pscostdown', 'pscostup', 'rootsolval', 'nNonz', and 'isBinary'. The function must output the 'score' and 'roundup', where 'score' is a float type indicating the variable's score, the more the better, and the 'roundup' is a bool type indicating whether we should round the variable up, True for rounding up. Be creative and do not give additional explanations. Designer idea: Allow weighting coefficients for each component.

*Reviewer* You are a code evaluator who tries to evaluate the code, giving relevant comments in the direction of code redundancy and code logic, etc.

*Judger* You are the final arbiter in the MA-Evolution System. Evaluate the heuristic code and review feedback to determine if it should be accepted, revised, or discarded. Output the final code and description.

## 8.8 GENERALIZATION FOR ACROSS PROBLEM CLASSES

To rigorously validate the generalization capabilities of DHEvo across different types of problem classes, we conducted an additional experiment. We constructed a composite dataset by mixing equal numbers of training instances from all four combinatorial problem classes (Setcover, Cauctions, Facilities, and Indset). We then applied the DHEvo framework to evolve a single heuristic algorithm on this highly heterogeneous mixture. The resulting heuristic was subsequently tested individually on each problem class.

Table 10: Performance comparison on different problem classes (Mean (Std)). The model was trained on a composite dataset and tested on individual tasks.

| Method | Cauctions (100) | Facilities | Setcover | Indset |
|---|---|---|---|---|
| **Ours** | **3.89 (2.19)** | **1.33 (0.89)** | 15.79 (13.96) | **1.17 (0.92)** |
| ReEvo | 7.76 (3.30) | 3.34 (1.99) | **8.01 (8.51)** | 14.26 (4.69) |
| LLM4Solver | 13.11 (12.96) | 1.29 (1.06) | 33.50 (27.53) | 13.32 (12.58) |
| EOH | 17.34 (22.32) | 2.51 (1.40) | 12.34 (11.19) | 1.29 (1.05) |

As shown in Table 10, our method achieves relatively balanced performance across all datasets. Unlike other methods that may fluctuate significantly (e.g., performing well on one task but poorly on another), our approach demonstrates the ability to generalize effectively to multiple problem classes without relying on domain-specific information.

## 8.9 HEURISTIC GENERATION FOR COMBINATORIAL OPTIMIZATION PROBLEMS

In this section, we evaluate the performance of our framework in generating heuristic strategies for well-known combinatorial optimization problems, including Online Bin Packing and the Traveling Salesman Problem.

**Online Bin Packing** The objective is to allocate a collection of items of different sizes into the fewest possible bins with a fixed capacity. We follow the settings in (Romera-Paredes et al., 2024)to design heuristics and use Weibull distribution instances with varying numbers of items (1k to 10k) and bin capacities (100 and 500). The performance of each method is measured by the fraction of excess bins used, where lower values indicate better performance. The task for DHEvo is to design the scoring function for assigning items. The inputs are the size of the item and the rest capacities of bins. The output is the scores for the bins.

Table 11: Online Bin Packing: Fraction of excess bins over the lower bound (mean, std) on Weibull-distributed instances

| Method | 1k_C100 | 1k_C500 | 5k_C100 | 5k_C500 | 10k_C100 | 10k_C500 | Overall |
|---|---|---|---|---|---|---|---|
| EOH | 2.47% (0.56%) | 1.33% (1.24%) | **1.17%** (1.04%) | 0.92% (1.00%) | **0.85%** (0.91%) | 0.73% (0.87%) | 0.73% (0.80%) |
| FunSearch | 3.29% (0.40%) | 1.79% (1.57%) | 2.03% (1.33%) | 1.59% (1.38%) | 1.74% (1.27%) | 1.50% (1.29%) | 1.50% (1.29%) |
| ReEvo | 3.27% (0.33%) | 1.76% (1.54%) | 2.03% (1.33%) | 1.59% (1.38%) | 1.76% (1.28%) | 1.51% (1.29%) | 1.50% (1.30%) |
| **Ours** | **2.13% (0.30%)** | **1.21% (1.01%)** | 1.25% **(0.83%)** | **1.00% (0.83%)** | 1.05% **(0.75%)** | **0.91% (0.76%)** | **0.91% (0.75%)** |

We evaluate the best heuristic discovered by our approach on instances of various sizes and capacities, comparing it against three representative LLM-based automatic algorithm design frameworks (i.e., EOH, FunSearch, and ReEvo). Table 11 presents the average gaps to the lower bounds, where the best results are highlighted in bold. Our method achieves superior results across all six tested scenarios. Notably, it consistently exhibits the lowest variance (standard deviation) on all problem instances. In terms of overall performance across all instances, our method achieves an average fraction of excess bins over the lower bound of 0.91% with a standard deviation of 0.75%.

**Traveling salesman Problem** The objective is to find the shortest possible route that visits a set of nodes exactly once and returns to the origin. We use DHEVO to design Guided Local Search (GLS) heuristics and evaluate them on instances with varying problem sizes (20, 50, 100, and 200 nodes). The performance of each method is measured by the relative distance to the best-known solutions, where lower values indicate better performance. The task for DHEVO is to design the heuristic function to guide the search process. The input is the distance matrix between nodes. The output is a matrix of prior indicators representing how unfavorable it is to include each edge in the solution.

We evaluate the best heuristic discovered by our approach on TSP instances of varying sizes, comparing it against three baseline methods (i.e., RE, EOH, and FunSearch). Table 12 presents the relative distance to the best-known solutions, where the best results are highlighted in bold. Our method achieves ideal results across all tested scenarios. Specifically, it attains a perfect 0.000% gap on instances with 20, 50, and 100 nodes, matching the theoretical best performance. Furthermore, on the largest instances (200 nodes), our method demonstrates superior scalability, achieving the

Table 12: Comparison of the relative distance (%) to the best-known solutions

| Method | TSP20 | TSP50 | TSP100 | TSP200 |
|--------|-------|-------|--------|--------|
| ReEvo | 0.000% | 0.000% | 0.000% | 0.216% |
| EOH | 0.000% | 0.000% | 0.000% | 0.216% |
| FunSearch | 0.000% | 0.000% | 0.104% | 0.647% |
| **Ours** | **0.000%** | **0.000%** | **0.000%** | **0.190%** |

lowest relative distance of 0.190%, outperforming EOH and ReEvo (0.216%) as well as FunSearch (0.647%).

## 8.10 COMPUTATIONAL COST ANALYSIS

We provide a detailed account of the training costs incurred when evolving heuristics on a single problem dataset. The evaluation considers three metrics: (1) *Compute Time* (total duration for training), (2) *Token Consumption* (total tokens processed), and (3) *Financial Cost* (estimated based on standard LLM API pricing, e.g., GPT-4o-mini).

Table 13 presents the comparison between our method and baseline frameworks. It is observed that our approach incurs higher time and token consumption ($\sim$124 min, $\sim$1,952k tokens) compared to EOH and ReEvo. This increase is attributed to the comprehensive multi-agent interactions and the detailed role-based prompts employed in our system. However, the absolute financial cost remains extremely low ($\approx$ \$0.029). We argue that this modest increase in training expense is well justified by the framework's superior generalization capabilities and robust performance across diverse problem classes.

Table 13: Comparison of computational costs for evolving heuristics on a single dataset. Costs are estimated based on GPT-4o-mini pricing models.

| Method | Time (min) | Token Consumption | Cost (USD) |
|--------|-----------|-------------------|------------|
| EOH | $\sim$60 | $\sim$141.75k | $\sim$\$0.00213 |
| ReEvo | $\sim$68 | $\sim$145.00k | $\sim$\$0.00218 |
| **Ours** | $\sim$124 | $\sim$1,952.00k | $\sim$\$0.02928 |

## 8.11 GENERATED HEURISTICS

This section presents the best heuristics generated by DHEvo.

```python
def myheurdiving(mayrounddown, mayroundup, candsfrac, candsol,
                nlocksdown, nlocksup, obj, objnorm,
                pscostdown, pscostup, rootsolval,
                nNonz, isBinary):
    score = 0.0
    roundup = False

    # Penalize if both rounding options are feasible
    if mayrounddown and mayroundup:
        score = -40

    # Evaluate candidate based on fractional part
    if candsfrac > 0.5:
        score += candsfrac * 80
        roundup = True
        if pscostup > 0.5:
            score += pscostup * 50
    else:
        score += (1 - candsfrac) * 60
        if pscostdown < -0.3:
            score -= abs(pscostdown) * 25
```

```
    # Normalize objective contribution
    score += (obj / (objnorm + 1e-6)) * 90

    # Adjust for locking counts
    score += (nlocksdown * 25 - nlocksup * 15)

    # Reward for non-zero entries and binary variable nature
    if nNonz > 2:
        score += nNonz * 20
    if isBinary:
        score += 50

    return score, roundup
```

Listing 1: Heuristic for cauctions

```
def myheurdiving(mayrounddown, mayroundup, candsfrac, candsol, nlocksdown
    , nlocksup, obj, objnorm, pscostdown, pscostup, rootsolval, nNonz,
    isBinary):
    score = 0.0
    roundup = False

    # Base score weighted by normalized objective contribution
    score += (obj/(objnorm + 1e-9)) * 5 if objnorm >0 else 0

    # Penalize rounding options to encourage exploration
    score -= nlocksdown * 7 if mayrounddown else 0
    score -= nlocksup *7 if mayroundup else 0

    # Favor large fractions away from 0.5 for exploration
    score += (abs(candsfrac - 0.5) * 10)

    # Adjust score based on solution value and its contribution
    score += (candsol / (1 + abs(rootsolval) * obj)) * 4 if rootsolval !=
        0 else 0

    # Employ pseudo costs to influence rounding decisions
    if pscostdown < 0 and mayrounddown:
        score += -pscostdown * 3 # Favor rounding down with negative pseudo
            costs
    if pscostup < 0 and mayroundup:
        score -= -pscostup * 3 # Discourage rounding up with negative
            pseudo costs

    #Determine rounding direction based on fractional part and exploration
        potential
    if candsfrac >= 0.7 and mayroundup:
        roundup = True
    elif candsfrac <= 0.3 and mayrounddown:
        roundup = False

    # Encourage solutions with fewer nonzero entries
    score += (1 / (nNonz + 1)) * 2 if nNonz > 0 else 0

    return score, roundup
```

Listing 2: Heuristic for facility

```
def myheurdiving(mayrounddown, mayroundup, candsfrac, candsol, nlocksdown
    , nlocksup, obj, objnorm, pscostdown, pscostup, rootsolval, nNonz,
    isBinary):
    score = 0.0
```

```
    # Strongly penalize feasible rounding options
    if mayrounddown:
        score -= 3.0
    if mayroundup:
        score -= -3.0

    # Incorporate fractional part and objective value
    score += (1.0 - candsfrac) * obj * 0.5 if mayrounddown else 0
    score += candsfrac * obj *0.5 if mayroundup else 0

    # Adjust with pseudo costs
    score += pscostdown * candsfrac * 1.5 if mayrounddown else 0
    score += pscostup * (1 - candsfrac) * 1.5 if mayroundup else 0

    # Apply less severe penalty for distance from the root solution
    score -= abs(rootsolval - candsol) * 0.1

    # Normalize the score
    if objnorm > 0 :
        score /= objnorm

    # Reward more for binary variables
    score += nlocksup * 0.3 - nlocksdown * 0.3
    if isBinary:
        score += 1.0

    # Determine rounding direction
    roundup = (score > 0) and (not isBinary or mayroundup)

    return score, roundup
```

Listing 3: Heuristic for indset

```
def myheurdiving(mayrounddown, mayroundup, candsfrac, candsol, nlocksdown
    , nlocksup, obj, objnorm, pscostdown, pscostup, rootsolval, nNonz,
    isBinary):
    score = 0.0

    # Strongly penalize feasible rounding options
    if mayrounddown:
        score -= 3.0
    if mayroundup:
        score -= -3.0

    # Incorporate fractional part and objective value
    score += (1.0 - candsfrac) * obj * 0.5 if mayrounddown else 0
    score += candsfrac * obj *0.5 if mayroundup else 0

    # Adjust with pseudo costs
    score += pscostdown * candsfrac * 1.5 if mayrounddown else 0
    score += pscostup * (1 - candsfrac) * 1.5 if mayroundup else 0

    # Apply less severe penalty for distance from the root solution
    score -= abs(rootsolval - candsol) * 0.1

    # Normalize the score
    if objnorm > 0 :
        score /= objnorm

    # Reward more for binary variables
    score += nlocksup * 0.3 - nlocksdown * 0.3
    if isBinary:
        score += 1.0

    # Determine rounding direction
```

```
    roundup = (score > 0) and (not isBinary or mayroundup)

    return score, roundup
```

Listing 4: Heuristic for indset

```python
def myheurdiving(mayrounddown, mayroundup, candsfrac, candsol, nlocksdown
    , nlocksup, obj, objnorm, pscostdown, pscostup, rootsolval, nNonz,
    isBinary):
    score = 0.0

    # Penalties for feasible rounding options to promote exploration
    if mayrounddown:
        score -= 10.0
    if mayroundup:
        score -= 10.0

    # Favor fractional values at extremes (0 or 1)
    score += (1 - abs(candsfrac - 0.5)) * 30.0

    # Normalize impact of the objective function
    score += (obj/(objnorm + 1e-5)) * 0.5

    #Include pseudo cost adjustments for better decision-making
    score += pscostup if mayroundup else 0.0
    score -= pscostdown if mayrounddown else 0.0

    # Integrate root solution value adjusted by variable complexity
    score += rootsolval / (nNonz + 1)

    # Amplify score for binary variables to encourage decisive rounding
    if isBinary:
        score *= 2.0

    # Determine rounding direction based on computed score and pseudo
        costs
    roundup = (mayrounddown and (pscostup <= pscostdown or not
        mayrounddown))

    return score, roundup
```

Listing 5: Heuristic for setcover

