# OpenReview forum: "DHEvo: Data-Algorithm Based Heuristic Evolution for Generalizable MILP Solving"
_ICLR.cc/2026/Conference — ICLR 2026 Conference Withdrawn Submission_

### Official Review · Reviewer_zwat · 2025-10-29

**Soundness:** 2
**Presentation:** 2
**Contribution:** 1
**Rating:** 4
**Confidence:** 4

**Summary:**

This paper presents DHEvo, a novel framework that co-evolves Mixed-Integer Linear Programming (MILP) problem instances and diving heuristics using a Large Language Model (LLM)-based multi-agent system. The core idea is to iteratively select "structurally representative" instances to guide the evolution of more generalizable heuristics, addressing the problem of high intra-class instance heterogeneity.

**Strengths:**

1. The concept of co-evolving both the data (MILP instances) and the algorithms (heuristics) is a creative contribution. Moving beyond a static training set to an adaptive, evolutionary selection of "representative" instances is a useful approach to improving generalization.
2. The experimental section is comprehensive. The method consistently outperforms a wide range of baselines on a lot of academic datasets and real-world problems.

**Weaknesses:**

1. The theoretical analysis in Section 3.2 is presented as a motivation. However, it does not explain why co-evolution on representative instances works; it only states that if you have a good set of base heuristics, combining them is safe. It does not address the core challenge of intra-class heterogeneity or explain how co-evolution helps.
2. The prompt engineering details in the appendix are excellent. However, it would be beneficial to include a brief discussion in the main text about how it is being designed and why this PE works.

**Questions:**

1. Can you demonstrate the generalization advantage of DHEvo more robustly, for example, by validating the generalization capabilities across different types of problem classes?

---

> ### Author Response · Authors · 2025-11-20
>
> We are grateful for the insightful and detailed feedback from the reviewer, which greatly strengthens our work. We provide our responses below and kindly refer the reviewer to the general response for broader discussions.
>
> ### **W1: Co-Evolution and Theoretical Motivation**
>
> Thank you for the comment. Section 3.2 is intended as a motivation, but it directly informs why co-evolution is needed. The theoretical results show that mixtures of atomic operators generalize as long as the training set is representative. However, due to intra-class heterogeneity (shown in Fig. 2), randomly sampled instances have high structural variance, making empirical risk minimization unstable and harming generalization.
>
> Co-evolution addresses precisely this gap, and it is a practical extension of this theoretical motivation: the evolutionary process consistently selects high-fitness instances, which our analysis and ablations show to be structurally simpler and more representative of the underlying class distribution. By iteratively refining both the instance set and the heuristics, co-evolution reduces variance in the training distribution and produces stable signals that satisfy the assumptions of the theoretical motivation.
>
> ### **W2: Prompt Engineering Strategy**
>
> Our prompts do not rely on any specially engineered mechanisms such as chain-of-thought, reflection, or self-debugging templates. Instead, they follow standard evolutionary operations similar to baseline prompt designs and thus remain broadly applicable to any heuristic discovery or LLM-based evolutionary computation task. We agree that providing a brief discussion of this in the main text would improve clarity.
>
> In particular, our PE strategy is intentionally streamlined to avoid unnecessary complexity while effectively supporting the co-evolution process.: (i) it assigns distinct roles within the multi-agent system (Designer/Coder/Reviewer/Judge) to structure generation, debugging, and refinement, and (ii) it incorporates evolution-specific objectives (e.g., mutation or crossover instructions) directly into prompts so the LLM produces code aligned with evolutionary operations. This lightweight PE is effective because it constrains the search space, reduces hallucination, and maintains consistency across generations, as supported by our ablation results.
>
>
> ### **Q1: Generalization Across Problem Classes**
>
> To rigorously validate the generalization capabilities of DHEvo across different types of problem classes, we conducted an additional experiment. We constructed a composite dataset by mixing equal numbers of training instances from all four combinatorial problem classes (Setcover, Cauctions, Facilities, and Indset). We then applied the DHEvo framework to evolve a single heuristic algorithm on this highly heterogeneous mixture, without providing the agent with any domain-specific labels.
>
> #### Performance Comparison
>
> | Method   | cauctions  | facilities       | Setcover         | Indset           |
> |----------|-----------------|------------------|------------------|------------------|
> | Ours     | mean=3.89, std=2.19 | mean=1.33, std=0.89 | mean=15.79, std=13.96 | mean=1.17, std=0.92 |
> | ReEvo    | mean=7.763, std=3.3 | mean=3.339, std=1.99 | mean=8.01, std=8.51  | mean=14.26, std=4.69 |
> | EA       | mean=13.11, std=12.96 | mean=1.29, std=1.06 | mean=33.50, std=27.53 | mean=13.32, std=12.58 |
> | EOH      | mean=17.34, std=22.32 | mean=2.51, std=1.40 | mean=12.34, std=11.19 | mean=1.29, std=1.05 |
>
> As shown in the table, our method achieves relatively balanced performance across all datasets, demonstrating its ability to generalize effectively to multiple problem classes without relying on domain-specific information.

---

> ### Author Response · Authors · 2025-11-27
> **Responses to Reviewers and Request for Feedback**
>
> Dear Reviewer  zwat：
>
> We hope the above clarifications and the additional experiments in the revised draft sufficiently addressed your concerns. If you are satisfied, we kindly request you to consider updating the score to reflect the newly added results and discussion. We remain committed to addressing any remaining points you may have during the discussion phase.
>
> Best,
>
> Authors

---

### Official Review · Reviewer_kx8b · 2025-10-30

**Soundness:** 3
**Presentation:** 3
**Contribution:** 2
**Rating:** 4
**Confidence:** 4

**Summary:**

This paper introduces DHEvo, a data-algorithm co-evolution framework that automatically generates generalizable diving heuristics for Mixed Integer Linear Programming (MILP) problems. DHEvo uses a multi-agent, LLM-driven evolutionary system to jointly evolve MILP instances and their heuristics. It identifies representative instance–heuristic pairs that generalize well within problem classes.  Experiments on synthetic benchmarks and real-world datasets show that DHEvo significantly outperforms state-of-the-art hand-crafted, learning-based, and LLM-based methods in solution quality (primal gap), variance, and solving efficiency.

**Strengths:**

- The idea of simultaneously evolving both instances and heuristics is well-motivated, justified, and proven effective through ablation studies.
- Using multiple LLM agents to generate, review, and select heuristics is methodologically sound. Ablation studies show this approach improves both solution quality and heuristic diversity.
- The experiments use synthetic and real-world datasets with clear performance metrics—such as primal gap and primal-dual integral—supporting the authors' claims through rigorous ablation studies.

**Weaknesses:**

- **Scope Limited to Diving Heuristics:** While the use case is well-motivated and relevant, DHEvo is applied only to diving heuristics. It remains unclear why the method targets this specific class of heuristics and whether it can extend to other MILP heuristics or broader combinatorial optimization problems. A discussion of limitations or necessary adaptations for other domains would strengthen the paper.
- **Computational and Resource Cost Reporting:** The paper lacks a fair comparison of computational resources, particularly the time and token consumption of LLM-based methods (including DHEvo and baselines). This information is essential for assessing practical deployment under real-world compute constraints.
- **Novelty with Respect to Prior Work:** The multi-agent, LLM-driven evolutionary search paradigm—including agent-based code design and review—has been explored in recent literature. The paper would benefit from clearly articulating which methodological and conceptual advances are novel to DHEvo.

**Questions:**

1. Is DHEvo in its current form specific to diving heuristics, or could it be extended to other primal heuristics, branching rules, or even non-MILP algorithms? What would need to be adapted for such generalization?
2. What is the average/total compute time, number of tokens, and any incurred API costs for running DHEvo compared to LLM-based baselines? This is important for practical adoption and reproducibility.

**Minor Comments**
- Eq. (2): Have you introduced and explained $\Delta^{k-1}$ before using it here?
- Line 366: Duplicated "Experimental setup"

---

> ### Author Response · Authors · 2025-11-20
>
> ---
>
> We are grateful for the insightful and detailed feedback from the reviewer, which greatly strengthens our work. We provide our responses below and kindly refer the reviewer to the general response for broader discussions.
>
> ---
>
> ### **1. Q1 & W1: Broader Applicability**
> The primary focus of this paper is to automatically design effective diving heuristics for MILP problems. However, our method is a general evolutionary computation framework that can also be applied to other types of heuristics. Our original plan was to extend this work to other CO problems in future research, and we have already evaluated the framework on multiple CO benchmarks. The results are summarized below and can be included in the appendix or main text to demonstrate the broader applicability of our approach.
>
> #### **Traveling Salesman Problem (TSP)**
> *Comparison of the relative distance (%) to the best-known solutions*
>
> | Method      | 20 nodes | 50 nodes | 100 nodes | 200 nodes |
> |-------------|----------|----------|-----------|-----------|
> | RE          | 0.000%   | 0.000%   | 0.000%    | 0.216%    |
> | EOH         | 0.000%   | 0.000%   | 0.000%    | 0.216%    |
> | FunSearch   | 0.000%   | 0.000%   | 0.104%    | 0.647%    |
> | **Ours**    | 0.000%   | 0.000%   | 0.000%    | **0.190%**|
> ---
>
> #### **Online Bin Packing**
> *Fraction of excess bins over the lower bound on Weibull-distributed instances (mean, std)*
>
> | Method      | 1k_C100           | 1k_C500            | 5k_C100            | 5k_C500            | 10k_C100           | 10k_C500           | Overall            |
> |-------------|-------------------|--------------------|--------------------|--------------------|--------------------|--------------------|--------------------|
> | EOH         | 2.47% (0.56%)     | 1.33% (1.24%)      | 1.17% (1.04%)      | 0.92% (1.00%)      | 0.85% (0.91%)      | 0.73% (0.87%)      | 0.73% (0.80%)      |
> | FunSearch   | 3.29% (0.40%)     | 1.79% (1.57%)      | 2.03% (1.33%)      | 1.59% (1.38%)      | 1.74% (1.27%)      | 1.50% (1.29%)      | 1.50% (1.29%)      |
> | ReEvo       | 3.27% (0.33%)     | 1.76% (1.54%)      | 2.03% (1.33%)      | 1.59% (1.38%)      | 1.76% (1.28%)      | 1.51% (1.29%)      | 1.50% (1.30%)      |
> | **Ours**    | **2.13% (0.30%)** | **1.21% (1.01%)**  | 1.25% (0.83%)      | **1.00% (0.83%)**  | 1.05% (0.75%)      | **0.91% (0.76%)**  | **0.91% (0.75%)**  |
>
> From the results on these two problems, we can see that our method consistently outperforms other baselines in terms of both average solution quality and stability across instances.
>
> ---
>
> ### **2. Q2 & W2: Training Cost**
> Our method is designed to generate a heuristic algorithm that generalizes across an entire problem class, rather than evolving on every individual instance encountered later. The cost of calling large language model (LLM) APIs is therefore considered training cost. In practical industrial scenarios, the benefits of a heuristic that can significantly accelerate solving are often much greater than the training cost. Similarly, one may not care the large training data and computation required by LLMs, as these are merely upfront investments; if the method delivers sufficient benefits during deployment, this investment is justified.
>
> For transparency and reproducibility, we report the training cost for DHEvo on a single dataset, including compute time, token usage, and estimated API expenses:
>
> | Method   | Time (min) | Token Consumption | Cost (USD) |
> |----------|------------|-------------------|------------|
> | EOH      | ~60        | ~141.75k          | ~\$0.00213 |
> | ReEvo    | ~68        | ~145k             | ~\$0.00218 |
> | **Ours** | **~124**   | **~1,952k**       | **~\$0.02928** |
>
> > **Note**: Costs are estimated based on standard LLM API pricing (e.g., GPT-4o-min or equivalent). Despite higher token usage, our framework’s improved generalization justifies the modest increase in training expense.
>
> ---

---

> ### Author Response · Authors · 2025-11-20
>
> ### **W3: Core Contribution and Novelty**
>
> DHEvo is not a prompt-engineering approach to enhance single-pass LLM code generation. Instead, our core contribution is a Data-Algorithm Co-Evolution Framework grounded in Evolutionary Computation theory, designed to tackle the fundamental challenge of generalization in heuristic design.
>
> 1. From Static Optimization to Co-Evolution
>    Existing LLM-based evolutionary methods (e.g., FunSearch, EOH) evolve heuristics on a fixed set of static instances. However, MILP instances within the same class often exhibit significant structural heterogeneity. Optimizing on a static or randomly sampled subset tends to produce heuristics that overfit and fail to generalize to unseen instances.
>
> 2. Theoretical Grounding for Generalization
>    In contrast to purely empirical heuristic-search works, DHEvo provides a theoretical guarantee for generalization. As shown in Section 3.2 and Theorem 2, using Rademacher Complexity, we prove that evolving heuristics on structurally representative instances—constructed as convex combinations of atomic operators—preserves tight generalization bounds.

---

> ### Author Response · Authors · 2025-11-27
> **Responses to Reviewers and Request for Feedback**
>
> Dear Reviewer kx8b:
>
> We hope the above clarifications and the additional experiments in the revised draft sufficiently addressed your concerns. If you are satisfied, we kindly request you to consider updating the score to reflect the newly added results and discussion. We remain committed to addressing any remaining points you may have during the discussion phase.
>
> Best,
>
> Authors

---

### Official Review · Reviewer_fvG3 · 2025-11-01

**Soundness:** 3
**Presentation:** 3
**Contribution:** 3
**Rating:** 6
**Confidence:** 5

**Summary:**

The paper proposes DHEvo, a data–algorithm co-evolution framework for generalizable MILP solving. It co-evolves representative instances and diving heuristics via an LLM-based multi-agent system (Designer/Coder/Reviewer/Judge), selecting high-fitness instance–heuristic pairs across generations, and integrates the learned heuristics into SCIP. Evaluations on four CO datasets (cauctions, setcover, facilities, indset) and three real-world settings (LoadBalance, NNVerify, MIPLIB) show consistent improvements in primal gap, solving time, and PDI over hand-crafted heuristics, learning baselines (L2DIVE), and LLM-based methods (LLM4Solver, FunSearch, EoH).

**Strengths:**

- Interesting and timely idea: jointly evolving data (instances) and algorithms (heuristics) to improve generalization within a problem class.
- Clear instantiation: multi-agent LLM pipeline, well-described evolutionary operators, and fitness-driven selection with temperature-based sampling.
- Strong empirical results: consistent gains over hand-crafted, learning-based, and LLM-based baselines; improvements carry over to full B&B with SCIP, and to real-world datasets.
- Ablations are helpful: show benefits of co-evolution, MA-Evolution, and data selection policies (simple/representative instances).

**Weaknesses:**

- Missing a key baseline: ReEvo is a prominent agent-based code-generation/evolution approach. Lack of comparison weakens the contribution claim, especially since DHEvo’s pipeline shares the agentic-evolution flavor.

- Benchmark breadth: classic LLM+EC works (e.g., EoH, FunSearch) commonly report on canonical CO tasks like Bin Packing and TSP. Not evaluating on such benchmarks limits comparability to the broader literature and makes it harder to assess transferability beyond MILP templates used here.


[1]Ye H, Wang J, Cao Z, et al. Reevo: Large language models as hyper-heuristics with reflective evolution[J]. Advances in neural information processing systems, 2024, 37: 43571-43608.

**Questions:**

- Will you add ReEvo as a baseline? At minimum, reproduce ReEvo’s agent framework on your four CO datasets with aligned prompts/evaluation to quantify the margin.

- Can you evaluate DHEvo on Bin Packing and TSP to match canonical CO reporting in EoH/FunSearch? Even a subset would strengthen external validity.

---

> ### Author Response · Authors · 2025-11-20
>
> ---
>
> We are grateful for the insightful and detailed feedback from the reviewer, which greatly strengthens our work. We provide our responses below and kindly refer the reviewer to the general response for broader discussions.
>
> ---
>
> ### **W1&Q1ReEvo as Baseline**
> Thank you for pointing out this omission. As a supplement, we have added ReEvo as a baseline. Its experimental results have been included below:
>
> | Method   | cauctions               | facilities              | Setcover                | Indset                 |
> |----------|-------------------------|-------------------------|-------------------------|------------------------|
> | LLM4Solver       | mean = 3.50, std = 2.50 | mean = 1.42, std = 0.85 | mean = 19.26, std = 18.33 | mean = 1.15, std = 1.00 |
> | Ours     | mean = 2.45, std = 1.96 | mean = 1.40, std = 0.70 | mean = 7.35, std = 9.74   | mean = 1.20, std = 1.07 |
> | ReEvo    | mean = 59.62, std = 6.11| mean = 1.21, std = 0.78 | mean = 8.43, std = 7.82   | mean = 1.05, std = 1.10 |
>
> > **Note:** All values represent the relative gap (%) to the optimal or best-known solution.
>
> ---
>
> ### **W2&Q2 :Benchmark Breadth**
>
> While this paper primarily focuses on the evolution of  diving heuristics for solving Mixed-Integer Linear Programming (MILP) problems, our method is in fact a general evolutionary computation framework. Our original plan was to extend this approach to other combinatorial optimization (CO) problems in future work. In fact, after submitting this paper, we have already evaluated our framework on multiple CO benchmarks. The results below demonstrate its broad applicability beyond MILP diving heuristics.
>
> ### 1. Traveling salesman problem
> *Comparison of the relative distance (%) to the best-known solutions*
>
> | Method      | 20 nodes | 50 nodes | 100 nodes | 200 nodes |
> |-------------|----------|----------|-----------|-----------|
> | RE          | 0.000%   | 0.000%   | 0.000%    | 0.216%    |
> | EOH         | 0.000%   | 0.000%   | 0.000%    | 0.216%    |
> | FunSearch   | 0.000%   | 0.000%   | 0.104%    | 0.647%    |
> | **Ours**    | **0.000%**  | **0.000%**   |**0.000%** | **0.190%**|
>
> ### 2.Online Bin Packing
> *Fraction of excess bins over the lower bound (mean, std) on Weibull-distributed instances*
>
> | Method      | 1k_C100           | 1k_C500            | 5k_C100            | 5k_C500            | 10k_C100           | 10k_C500           | Overall            |
> |-------------|-------------------|--------------------|--------------------|--------------------|--------------------|--------------------|--------------------|
> | EOH         | 2.47% (0.56%)     | 1.33% (1.24%)      | 1.17% (1.04%)      | 0.92% (1.00%)      | 0.85% (0.91%)      | 0.73% (0.87%)      | 0.73% (0.80%)      |
> | FunSearch   | 3.29% (0.40%)     | 1.79% (1.57%)      | 2.03% (1.33%)      | 1.59% (1.38%)      | 1.74% (1.27%)      | 1.50% (1.29%)      | 1.50% (1.29%)      |
> | ReEvo       | 3.27% (0.33%)     | 1.76% (1.54%)      | 2.03% (1.33%)      | 1.59% (1.38%)      | 1.76% (1.28%)      | 1.51% (1.29%)      | 1.50% (1.30%)      |
> | **Ours**    | **2.13% (0.30%)** | **1.21% (1.01%)**  | 1.25% (0.83%)      | **1.00% (0.83%)**  | 1.05% (0.75%)      | **0.91% (0.76%)**  | **0.91% (0.75%)**  |
>
> From the results on these two problems, we can see that our method consistently outperforms other baselines in terms of both average solution quality and stability across instances. Specifically, for TSP, our method achieves the smallest relative gap on the largest instances, and for Online Bin Packing, it achieves lower or comparable excess bins with reduced variance, demonstrating its generalization capability and robustness across different CO problem types.

---

> ### Author Response · Authors · 2025-11-27
> **Responses to Reviewers and Request for Feedback**
>
> Dear Reviewer fvG3:
>
> As the discussion phase is approaching its end, we kindly request the reviewer to let us know if the above clarifications and the previously added experiments have addressed the remaining questions. We would be happy to address any additional points the reviewer may have during the remaining time of the discussion phase.
> We thank the reviewer for engaging with us in the discussion.
>
> Best,
>
> Authors

---

### Official Review · Reviewer_MQcE · 2025-11-01

**Soundness:** 3
**Presentation:** 3
**Contribution:** 3
**Rating:** 4
**Confidence:** 3

**Summary:**

The paper proposes DHEvo, a framework that co-evolves the heuristic function and the data in MILP solving, aiming to obtain heuristic functions with better generalization across different types of problems. The proposed approach demonstrates superior performance compared to several baseline methods.

**Strengths:**

- The paper provides a solid theoretical motivation, offering guidance for the joint evolution of heuristics and data.
- The experiments include ablation studies on the main components of DHEvo, showing a well-structured evaluation.

**Weaknesses:**

Several parts of the paper are unclear or insufficiently explained; see the questions below.

**Questions:**

1. The key idea of DHEvo is the “co-evolution of data and heuristics.” However, according to *Algorithm 2: DHEvo Framework*, it seems that DHEvo does not actually synthesize new data but rather modifies the evaluation data during heuristic evolution. Could the authors clarify this point?
2. Could the authors compare the experimental results in Tables 1, 3, and 4 with those obtained using Gurobi?
3. In the appendix, the authors mention that 20 MILP instances were selected from MIPLIB for evaluation. Since MIPLIB contains hundreds of instances, what criteria were used to select these 20? Could the authors evaluate DHEvo on additional MIPLIB instances?
4. During the evolution and evaluation phases of DHEvo, are the same data used? In other words, did the experiments distinguish between a “training set” (used for heuristic evolution) and a test set?
5. The paper claims that DHEvo addresses the issue of “yielding heuristics that lack generalization” in existing LLM-based methods. Which experiment specifically evaluates the generalization ability of the heuristics generated by DHEvo?

---

> ### Author Response · Authors · 2025-11-20
>
> ---
> We are grateful for the insightful and detailed feedback from the reviewer, which greatly strengthens our work. We provide our responses below and kindly refer the reviewer to the general response for broader discussions.
>
> ---
>
>
> ### **Q1: Clarification on the Core Idea of DHEvo**
> We would like to clarify that the core idea of DHEvo is indeed the co-evolution of data and heuristics, but this does not imply the generation of entirely new instances. Instead, our method iteratively selects the most representative instances from the training set and performs evolutionary computation on these instances to obtain heuristics with strong generalization ability. Correspondingly, during the training process, the instances in the selected dataset are continuously refined and updated, and the heuristics are evolved in tandem with them. In this way, DHEvo realizes a synchronized co-evolution of heuristics and instances, without synthesizing entirely new data.
>
>
> ---
>
> ### **Q2: Comparison with Gurobi**
> Thank you for the suggestion. Gurobi is a closed-source commercial solver, and there exists a substantial performance gap between SCIP and Gurobi. Therefore, directly comparing our method (built on SCIP) with Gurobi on the results in Tables 1, 3, and 4 would not be fair. To mitigate the impact of solver-specific differences, we focused our comparison on the first node of the search tree, where the heuristic code has the most significant influence, as shown in Table 2 of the paper. This allows for a more meaningful evaluation of the heuristics themselves, independent of the solver’s inherent capabilities.
>
> ---
>
> ### **Q3: Evaluation on MIPLIB**
> MIPLIB is a heterogeneous benchmark containing instances from many different problem classes, with highly diverse internal distributions. The common practice, as in [1], is to train and evaluate algorithms separately on each problem class. Therefore, testing a single generated algorithm across the entire MIPLIB is not appropriate. Moreover, many instances in MIPLIB are unsuitable for diving-based heuristics. For example, in certain supply chain instances, diving algorithms cannot find feasible solutions, or the solver does not invoke diving heuristics at all. Therefore, even the most effective diving algorithm would be meaningless on such problems.
>
> In our experiments, we selected the subset of instances where diving heuristics could actually take effect, which resulted in the 20 instances reported in the paper.
>
> [1].Learning Cut Selection for Mixed-Integer Linear Programming via Hierarchical Sequence Model
>
> ---
>
> ### **Q4: Training vs. Testing Instances**
> Yes, to ensure the evolved heuristics have strong generalization ability, we do not evolve and evaluate heuristics on the same instances. Our method explicitly distinguishes between a training set and a test set. For example, on the cactions dataset, we randomly generated 500 instances, selected 50 instances for training, and used 100 instances for testing.
>
> ---
>
> ### **Q5: Measuring Generalization Ability**
> To evaluate the generalization ability of heuristics generated by DHEvo, we measure the variance of the relative gaps to the optimal values across instances within a problem class. As shown in Tables 1, 2, 3, and 5, heuristics produced by our method exhibit lower variance compared to other evolutionary approaches, indicating superior generalization performance.
>
> ---

---

> ### Author Response · Authors · 2025-11-27
> **Responses to Reviewers and Request for Feedback**
>
> Dear Reviewer MQcE:
>
> We hope the above clarifications  sufficiently addressed your concerns. If you are satisfied, we kindly request you to consider updating the score to reflect the newly added results and discussion. We remain committed to addressing any remaining points you may have during the discussion phase.
>
> Best,
>
> Authors

---

### Author Response · Authors · 2025-11-20
**General Response to All Reviewers**

We thank all reviewers for their time and effort in reviewing the paper. We are delighted to see that all the reviewers appreciated our work and provided valuable feedback. Here we address several common questions raised by multiple reviewers, while deferring reviewer-specific comments to individual responses. Based on your feedback, we have conducted the following additional experiments and analyses to further strengthen our paper:

- **[GR1]** We added experiments on other combinatorial optimization and operations research problems to validate the generalization and applicability of our method beyond MILP diving heuristics.

- **[GR2]** We provide a detailed report on the training cost incurred when evolving heuristics on a single problem dataset, including compute time, token consumption, and estimated API expenses.

- **[GR3]** We further clarify the design details of our method and its connection to the theoretical analysis, and we include additional baselines to enable a more comprehensive empirical evaluation.

- **[GR4]** We conducted a cross-dataset experiment to evaluate the generalization ability of our method across heterogeneous problem instances, without access to domain-specific labels.

---

> ### Author Response · Authors · 2025-11-20
> **GR1:Expand to other optimization problems**
>
> Some of you asked about the applicability of DHEvo beyond MILP diving heuristics and whether it can generalize to other combinatorial optimization problems. To address this, we conducted additional experiments on the Traveling Salesman Problem  and Online Bin Packing. We tested the best heuristic produced by our method and compared it with other LLM-based evolutionary computation methods.
>
> For Online Bin Packing, the problem sizes range from 1k to 10k items, with capacities of 100 and 500. The table below shows the average gaps to the lower bounds. Our method achieves strong performance across multiple instance categories, demonstrating its generalization capability. For the Traveling Salesman Problem , we consider instances with 20, 50, and 100 locations. The heuristics generated by our method are embedded into a Guided Local Search (GLS) algorithm. Table 3 shows the average performance of the heuristics on these random instances. Our method consistently achieves competitive results across all instance sizes.
> ### 1. Traveling salesman problem
> *Comparison of the relative distance (%) to the best-known solutions*
>
> | Method      | 20 nodes | 50 nodes | 100 nodes | 200 nodes |
> |-------------|----------|----------|-----------|-----------|
> | RE          | 0.000%   | 0.000%   | 0.000%    | 0.216%    |
> | EOH         | 0.000%   | 0.000%   | 0.000%    | 0.216%    |
> | FunSearch   | 0.000%   | 0.000%   | 0.104%    | 0.647%    |
> | **Ours**    | **0.000%**  | **0.000%**   |**0.000%** | **0.190%**|
>
> ### 2.Online Bin Packing
> *Fraction of excess bins over the lower bound (mean, std) on Weibull-distributed instances*
>
> | Method      | 1k_C100           | 1k_C500            | 5k_C100            | 5k_C500            | 10k_C100           | 10k_C500           | Overall            |
> |-------------|-------------------|--------------------|--------------------|--------------------|--------------------|--------------------|--------------------|
> | EOH         | 2.47% (0.56%)     | 1.33% (1.24%)      | 1.17% (1.04%)      | 0.92% (1.00%)      | 0.85% (0.91%)      | 0.73% (0.87%)      | 0.73% (0.80%)      |
> | FunSearch   | 3.29% (0.40%)     | 1.79% (1.57%)      | 2.03% (1.33%)      | 1.59% (1.38%)      | 1.74% (1.27%)      | 1.50% (1.29%)      | 1.50% (1.29%)      |
> | ReEvo       | 3.27% (0.33%)     | 1.76% (1.54%)      | 2.03% (1.33%)      | 1.59% (1.38%)      | 1.76% (1.28%)      | 1.51% (1.29%)      | 1.50% (1.30%)      |
> | **Ours**    | **2.13% (0.30%)** | **1.21% (1.01%)**  | 1.25% (0.83%)      | **1.00% (0.83%)**  | 1.05% (0.75%)      | **0.91% (0.76%)**  | **0.91% (0.75%)**  |
>
> From the results on these two problems, we can see that our method consistently outperforms other baselines in terms of both average solution quality and stability across instances. Specifically, for TSP, our method achieves the smallest relative gap on the largest instances, and for Online Bin Packing, it achieves lower or comparable excess bins with reduced variance, demonstrating its generalization capability and robustness across different CO problem types.

---

> ### Author Response · Authors · 2025-11-20
> **GR2：Computational and Resource Cost Reporting(Training cost)**
>
> We provide a detailed account of the training cost incurred when evolving heuristics on a single problem dataset. This includes:
>
> Compute time: Total and average time required for training the heuristic algorithm.
>
> Number of tokens: Total tokens processed through the LLM during the evolutionary process.
>
> API usage and costs: Any calls to the LLM API, along with the associated monetary cost.
>
> | Method   | Time (min) | Token Consumption | Cost (USD) |
> |----------|------------|-------------------|------------|
> | EOH      | ~60        | ~141.75k          | ~\$0.00213 |
> | ReEvo    | ~68        | ~145k             | ~\$0.00218 |
> | **Ours** | **~124**   | **~1,952k**       | **~\$0.02928** |
>
> > **Note**: Costs are estimated based on standard LLM API pricing (e.g., GPT-4o-min or equivalent). Despite higher token usage, our framework’s improved generalization justifies the modest increase in training expense.

---

> ### Author Response · Authors · 2025-11-20
> **GR3:Methodological Clarifications, Theoretical Connections, and Additional Baselines**
>
> We have revised the manuscript to provide a deeper explanation of the DHEvo framework, specifically clarifying the mechanism of "Data-Algorithm Co-Evolution" and its intrinsic link to our theoretical analysis. Additionally, we have conducted new experiments for diving.
>
> DHEvo maintains a dynamic pool of instances. In each generation, the framework selects "representative" instances (those with high fitness/compatibility) to form the training environment for the next cycle.This creates a bidirectional feedback loop: the algorithms evolve to solve the current instances, while the instance population "evolves" to filter out noise and retain only those samples that effectively guide the search toward valid structural patterns.
>
> Our co-evolutionary framework is a practical extension designed to operationalize the theoretical motivation in Section 3.2. While Theorem 2 guarantees that mixtures of atomic operators generalize, this bound relies on the assumption that the empirical risk is minimized on a representative training set. However, due to the intra-class heterogeneity visualized in Figure 2, a randomly sampled training set exhibits high structural variance. Optimizing on such a noisy distribution is unstable, as the solver struggles to distinguish essential structural rules from instance-specific complexities. Co-evolution bridges this gap by functioning as a variance-reduction mechanism. By iteratively selecting high-fitness instances, which our ablation studies confirm are structurally simpler and "cleaner". This effectively filters out structural noise and produces stable learning signals, thereby ensuring the assumptions required for the theoretical generalization bounds are met in practice.
> In addition，we have added ReEvo as a baseline. Its experimental results have been included below:
>
> | Method   | cauctions               | facilities              | Setcover                | Indset                 |
> |----------|-------------------------|-------------------------|-------------------------|------------------------|
> | LLM4Solver| mean = 3.50, std = 2.50 | mean = 1.42, std = 0.85 | mean = 19.26, std = 18.33 | mean = 1.15, std = 1.13 |
> | ReEvo    | mean = 59.62, std = 6.11| mean = 1.21, std = 0.78 | mean = 8.43, std = 7.82   | mean = 1.05, std = 1.10 |
> | EoH | mean = 3.15, std = 3.15| mean = 1.47, std = 0.80 | mean = 19.70, std = 20.39   | mean = 1.06, std = 0.92  |
> | Ours     | mean = 2.45, std = 1.92 | mean = 1.40, std = 0.70 | mean = 7.35, std = 9.74   | mean = 1.07, std = 1.20 |
> > **Note:** All values represent the relative gap (%) to the optimal or best-known solution.

---

> ### Author Response · Authors · 2025-11-20
> **GR4: Generalization capabilities across different types of problem classes**
>
> To rigorously validate the generalization capabilities of DHEvo across different types of problem classes, we conducted an additional experiment. We constructed a composite dataset by mixing equal numbers of training instances from all four combinatorial problem classes (Setcover, Cauctions, Facilities, and Indset). We then applied the DHEvo framework to evolve a single heuristic algorithm on this highly heterogeneous mixture. The resulting heuristic was subsequently tested individually on each problem class.
>
> #### Performance Comparison
>
> | Method   | cauctions (100) | facilities       | Setcover         | Indset           |
> |----------|-----------------|------------------|------------------|------------------|
> | Ours     | mean=3.89, std=2.19 | mean=1.33, std=0.89 | mean=15.79, std=13.96 | mean=1.17, std=0.92 |
> | ReEvo    | mean=7.763, std=3.3 | mean=3.339, std=1.99 | mean=8.01, std=8.51  | mean=-14.26, std=4.69 |
> | LLM4Slover      | mean=13.11, std=12.96 | mean=1.29, std=1.06 | mean=33.50, std=27.53 | mean=13.32, std=12.58 |
> | EOH      | mean=17.34, std=22.32 | mean=2.51, std=1.40 | mean=12.34, std=11.19 | mean=1.29, std=1.05 |
>
>
> As shown in the table, our method achieves relatively balanced performance across all datasets, demonstrating its ability to generalize effectively to multiple problem classes without relying on domain-specific information.

---

### Author Response · Authors · 2025-11-26
**General Response: Our Revised Manuscript**

We have revised the paper to incorporate the results and discussions presented in the rebuttal. We colored the updates in blue. Specifically, we have made the following changes:

| Addition | Location in the Paper |
| :--- | :--- |
| Comparison on Combinatorial Problems | Appendix 8.9 |
| Cross-Problem Generalization Analysis| Appendix 8.8|
| Prompt Engineering Analysis  |Sec 3.4|
| Computational Cost Analysis | Appendix 8.10 |
|Reviewer kx8b: Minor Comments | We have incorporated these revisions into the corresponding sections of the manuscript. |

---

### Note · Authors · 2026-03-24

I have read and agree with the venue's withdrawal policy on behalf of myself and my co-authors.

---

### Meta-Review · Area_Chair_2dj3 · 2026-01-08

**Summary:**

The paper proposes DHEvo, a data-algorithm co-evolution framework utilizing Large Language Models to automatically generate generalizable diving heuristics for MILP solvers. The reviewers provided extensive feedback, and while the authors made substantial efforts in the rebuttal, key concerns remain unaddressed and the responses have introduced potential contradictions. First, the refusal to fully compare with Gurobi exposes a critical limitation: the method relies on modifying the solver's source code and recompilation to access internal features, making it impractical for commercial "plug-and-play" applications. Second, a major contradiction arises regarding generalization; while the paper explicitly claims the ability to handle heterogeneous problems (within or across classes), the authors are not able to test on MIPLIB specifically because of its diversity. Finally, the evidence for broad applicability to general Combinatorial Optimization (e.g., TSP) is ambiguous, as it is unclear whether these results utilize the proposed diving heuristics—which would contradict the exclusion of diverse MILP benchmarks—or require substantial, undisclosed adaptations. Therefore, the paper is not ready to publish at this time.

**Reviewer Concerns:**

### [MQcE]

Q2: The concern regarding the Gurobi comparison is not fully addressed. Since Gurobi supports user-defined heuristics via callbacks, a direct comparison is not inherently "unfair." The true bottleneck appears to be the method's implementation strategy: it relies on modifying the solver's source code to inject a C-Python interface and recompiling the solver to access specific internal features. This reveals a significant limitation: the proposed framework is not "plug-and-play" and cannot be easily adapted to powerful commercial solvers (which are closed-source but provide API/callback), thereby restricting its practical applicability.

Additionally, the caption of Table 2 should explicitly specify that the reported results are restricted to the root node evaluation.

Q3: The authors' rebuttal contradicts the claims made in the paper. The paper explicitly positions the method as a solution for "generalizable MILP solving" and emphasizes its ability to handle "substantial structural and distributional heterogeneity". Furthermore, in Section 8.8, the authors conduct a specific experiment to validate generalization across problem classes and claim that DHEvo "demonstrates the ability to generalize effectively to multiple problem classes without relying on domain-specific information." However, the rebuttal argues that testing on the broader MIPLIB is inappropriate precisely because it is a "heterogeneous benchmark" with "highly diverse internal distributions." Therefore, it's unclear whether the proposed method works well for heterogeneous datasets.

Other concerns are addressed.

### [fvG3]

All concerns are addressed.

### [kx8b]

W1 & Q1: Not fully addressed. The rebuttal claims broad applicability beyond MILP diving, but it remains unclear how DHEvo transfers to general CO settings and what adaptations are required. If the extension is simply “formulate CO as MILP and apply the evolved diving heuristic,” then the method would seemingly apply broadly—yet this contradicts the earlier rebuttal that diving heuristics are only meaningful on a limited subset of diverse benchmarks (e.g., MIPLIB). Conversely, if achieving the reported TSP/bin-packing results requires substantial changes (new representations, operators, evaluation loops, or solver integration), then these experiments do not directly substantiate the paper’s claim that the current method is broadly applicable; they instead suggest a different method family that needs to be specified.

Others are addressed.

### [zwat]

Q1: Not fully addressed. See comments before.

Other concerns are addressed.

**Reviewer Scores:**

MQcE is likely to maintain a 4 or potentially downgrade to 2, as multiple concerns remain unaddressed.

fvG3 is likely to hold at 6; while their specific concerns were resolved, the broader critical context makes a score increase unlikely.

kx8b and zwat are likely to maintain a score of 4, as the rebuttal did not fully resolve all raised issues.

---

### Decision · Program_Chairs · 2026-01-26

Reject